# DELVING INTO THE OPENNESS OF CLIP

## ABSTRACT

Contrastive Language-Image Pre-training (CLIP) has demonstrated great potential in realizing open-vocabulary visual recognition in a matching style, due to its holistic use of natural language supervision that covers unconstrained real-world visual concepts. However, it is, in turn, also difficult to evaluate and analyze the openness of CLIP-like models, since they are in theory open to any vocabulary but the actual accuracy varies. To address the insufficiency of conventional studies on openness, we resort to an incremental perspective and define the *extensibility*, which essentially approximates the model's ability to deal with new visual concepts, by evaluating openness through vocabulary expansions. Our evaluation based on extensibility shows that CLIP-like models are hardly truly open and their performances degrade as the vocabulary expands to different degrees. Further analysis reveals that the over-estimation of openness is not because CLIP-like models fail to capture the general similarity of image and text features of novel visual concepts, but because of the confusion among competing text features, that is, they are not *stable* with respect to the vocabulary. In light of this, we propose to improve the openness of CLIP in feature space by enforcing the distinguishability of text features. Our method retrieves relevant texts from the pre-training corpus to enhance prompts for inference, which boosts the extensibility and stability of CLIP even without fine-tuning.

## 1 INTRODUCTION

The seek for an intrinsically open mechanism of visual recognition (Deng et al., 2009; He et al., 2016) has always been a shared goal in the computer vision community (Scheirer et al., 2013; Geng et al., 2021; Bendale & Boult, 2015). It requires models to maintain flexibility to cope with the scaling of the recognition target, where both input images and the corresponding classes will dynamically expand according to actual needs. For example, in medical diagnosis (Razzak et al., 2017), new diseases emerge constantly and in e-commerce, new categories of products appear daily (Xu et al., 2019), which cannot be predefined in a finite class set and remain fixed during inference.

Faced with this challenging *open-world recognition* problem, traditional supervised classifiers and algorithms have struggled, as they only learn to discriminate limited classes in a closed set, and cannot adapt to the scaling of target classes. However, the emergence of Contrastive Language-Image Pre-training (CLIP) (Radford et al., 2021) and its *open-vocabulary learning* paradigm has reversed this situation. CLIP models visual recognition as a task of image-text matching rather than the classic image classification. It is pre-trained on web-scale collections of image-text pairs, learning unconstrained visual concepts from natural language supervision with contrastive learning. During inference, it devises a textual prompt like "a photo of a [CLASS]", where the class token can be replaced by any potential class name from a vocabulary. The prompt-formed class description with the highest similarity to the input image is predicted as the target class. This modeling paradigm makes CLIP *operationally* suitable for open tasks in the real world. When input images and the target classes change, CLIP can still conduct zero-shot inference by adaptively adjusting the class names in the vocabulary and then modifying the corresponding class descriptions for matching, sparing re-training the entire model on new data like the traditional classification-based methods. Nevertheless, contrary to the note "*CLIP has a wide range of capabilities due to its ability to carry out arbitrary image classification tasks*" in (Radford et al., 2021), previous evaluation of CLIP is still limited in the closed set, leaving its actual performance on open tasks in shadow.

In this work, we rethink openness, the intriguing but under-explored property of CLIP, and present a protocol for evaluating the openness of CLIP-like models (Radford et al., 2021; Li et al., 2021b; Mu et al., 2021; Yao et al., 2021; Zhou et al., 2021) from an incremental view. Specifically, we define *extensibility*, which essentially approximates the models' ability in dealing with new visual concepts through vocabulary expansion. Our experimental results based on extensibility show that CLIP and its variants have a significant drop in accuracy, e.g., $12.9\%$ of CLIP (RN101) on CIFAR100 as the vocabulary size expands from 5 to 100, indicating that the limited zero-shot capability of CLIP-like models is not sufficient to support its deployment in the open world. Different from previous openness-related work, we focus on analyzing how the new class descriptions introduced with vocabulary expansion affect the *stability* of classification on the old input images. Our investigation reveals that the small margin between text features of different classes leads to the prediction shift. To improve the distinguishability of text features and the semantic alignment between images and their textual description, we propose a non-parametric method named Retrieval-enhanced Prompt Engineering (REPE), which retrieves relevant captions from the pre-training corpus to customize the prompt for each class during zero-shot inference.

To summarize, our contribution is three-fold: **(1)** To our best knowledge, we are the first to systematically investigate the openness of CLIP, for which we design the evaluation protocol and two indicators of extensibility and stability. Through an analysis of the prediction shift during vocabulary expansion, we find that the performance of CLIP is greatly reduced by adding a small number of adversarial non-target classes, exposing the huge risk of its deployment in the open world. **(2)** We further dissect the feature space of CLIP from the perspectives of representation alignment and uniformity, observing that the uniformity of the textual space is critical for better extensibility. **(3)** We propose a simple yet effective method, REPE, to improve the extensibility and stability of CLIP without fine-tuning.

## 2 OPENNESS, EXTENSIBILITY, AND STABILITY

In this section, we first review CLIP's visual recognition paradigm based on image-text matching, and then demonstrate how it realizes open-vocabulary image classification in theory by vocabulary expansion (§ 2.1). To quantify the actual performance of CLIP-like models as the vocabulary expands, we define the metric of extensibility and propose a systematical evaluation protocol (§ 2.2). The experimental results and further analysis reveal that, as the vocabulary expands, the predictions of CLIP are unstable and prone to drift to the competing class descriptions that are newly introduced, which limits its extensibility and leaves a huge security risk when deployed in real-world applications (§ 2.3).

### 2.1 OPENNESS OF CLIP

In contrast to the classic supervised methods (He et al., 2016; Dosovitskiy et al., 2021), CLIP (Radford et al., 2021) models visual recognition as an image-text matching task with self-supervised contrastive pre-training. Formally, let $f$ be the CLIP model, it takes an image $\mathbf{x}$ and a *target vocabulary* $\mathcal{V}^{(T)} = \{w_i\}$ of the class names $w_i$ as inputs, and outputs the predicted label $\hat{y}$ of the image as:

$$\hat{y} = f\left(\mathbf{x}, \mathcal{V}^{(T)}\right) = \arg\max_i P\left(y = i \mid \mathbf{x}\right)$$

$$= \arg\max_i \frac{\exp\left(\text{sim}(f_T(\mathbf{t}_i), f_I(\mathbf{x}))\right)}{\sum_{j=1}^{|\mathcal{V}^{(T)}|} \exp\left(\text{sim}(f_T(\mathbf{t}_j), f_I(\mathbf{x}))\right)}, \quad (1)$$

where $\mathbf{t}_i$ is the textual description of the class name $w_i$ in a prompt format, e.g., "`a photo of a` $w_i$", $\text{sim}(\cdot, \cdot)$ denotes cosine similarity, $f_T$ and $f_I$ is the text and image encoder in CLIP, respectively. Such a modeling paradigm can realize the open-world image classification in theory by extending the target vocabulary $\mathcal{V}^{(T)}$ to arbitrary degrees. However, in most previous work (Radford et al., 2021; Li et al., 2021b; Mu et al., 2021; Yao et al., 2021; Zhou et al., 2021), CLIP is evaluated with a fixed $\mathcal{V}^{(T)}$ depending on the target classes of the downstream dataset $\mathcal{D}^{(T)}$:

$$\text{Acc}\left(\mathcal{V}^{(T)}\right) = \frac{1}{|\mathcal{D}^{(T)}|} \sum_{(\mathbf{x}, y) \in \mathcal{D}^{(T)}} \mathbb{I}\left(f\left(\mathbf{x}, \mathcal{V}^{(T)}\right) = y\right), \quad (2)$$

where $|\mathcal{D}^{(T)}|$ denotes the size of the dataset, and $\mathbb{I}(\cdot)$ is the indicator function. This vanilla evaluation setting with restricted input images and classes is insufficient for the open recognition tasks, as it

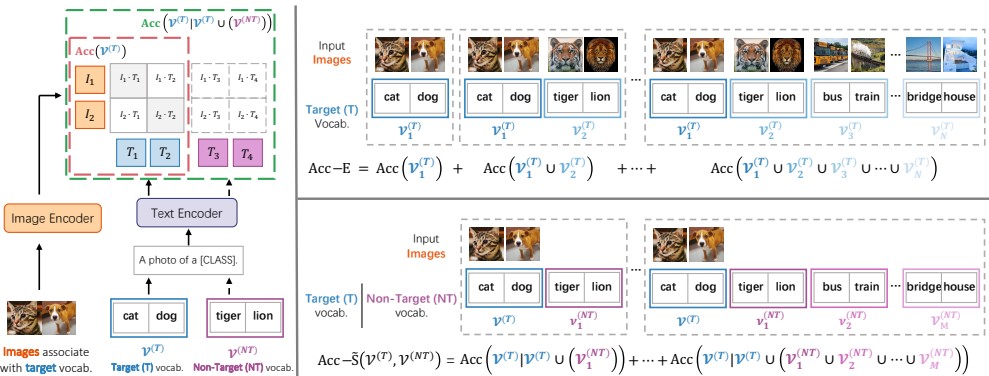

Figure 1: **Left**: the original accuracy of CLIP with target vocabulary (Eq.2) and the conditional accuracy of CLIP with non-target vocabulary (Eq. 5). In the latter, the classes from the non-target vocabulary are involved as distractors for the input images restricted in the target vocabulary. **Upper right**: calculation of Acc-E (Eq. 3). It measures the extensibility of models when recognition targets including both classes and the associated input images are scaling simultaneously. **Bottom right**: calculation of Acc-S (Eq. 6), which is a sub-problem introduced by Acc-E. It measures the prediction stability on the images from the target vocabulary as the distractors from the non-target vocabularies are incorporated incrementally.

does not explicitly and systematically models the dynamics of vocabulary expansion during inference, and thus cannot reflect the actual openness of CLIP in the face of real-life class scaling.

## 2.2 QUANTIFYING EXTENSIBILITY OF CLIP FOR THE OPEN WORLD

To quantify the model's capability in dealing with newly emerged recognition targets, we propose an evaluation protocol and define a metric of extensibility based on vocabulary expansion. Concretely, we incrementally expand the vocabulary $\mathcal{V}^{(T)}$ in Eq. 2 by introducing new classes and the associated input images, then evaluate the accuracy after each vocabulary expansion. These values of accuracy measure the dynamic performance of the model as the openness gradually increases, and their expected average is defined as the model's extensibility. In practice, we achieve this expansion by incrementally unioning $N$ disjoint target vocabularies [1] as shown in the upper right panel of Figure 1.

**Definition 2.1** (Extensibility). Given $N$ disjoint target vocabularies $\{\mathcal{V}_1^{(T)}, \cdots, \mathcal{V}_N^{(T)}\}$, we denote $\mathcal{S}_N$ as their full permutation and $\mathcal{V}_{s_i}^{(T)}$ as the $i^{(th)}$ vocabulary in a permutation $s \in \mathcal{S}_N$. When we union the $i^{(th)}$ vocabulary with the previous $i-1$ vocabularies, we achieve a vocabulary expansion and obtain $\mathcal{V}_{s_1}^{(T)} \cup \cdots \cup \mathcal{V}_{s_i}^{(T)}$. The extensibility refers to the averaged classification accuracy across $N$ incremental expansions as $i$ increases from 1 to $N$:

$$\text{Acc-E} = \underset{s \in \mathcal{S}_N}{\mathbb{E}} \frac{1}{N} \sum_{i=1}^{N} \text{Acc}\left(\mathcal{V}_{s_1}^{(T)} \cup \cdots \cup \mathcal{V}_{s_i}^{(T)}\right). \tag{3}$$

**Experimental settings** We evaluate the extensibility of CLIP and its variants, including DeCLIP (Li et al., 2021b), SLIP (Mu et al., 2021), Prompt Ensemble (Radford et al., 2021), CoOp (Zhou et al., 2021), on the CIFAR100 (Krizhevsky & Hinton, 2009) and ImageNet (Deng et al., 2009) datasets. Non-matching methods (Gao et al., 2021; Zhang et al., 2021; Wortsman et al., 2021) like linear probing, etc., are NOT included since they require training a classifier with *finite* class vectors, and thus are not suitable for class scaling in operation. To construct the vocabulary, we leverage the underlying superclass-class hierarchical structure of the two datasets (Krizhevsky & Hinton, 2009; Santurkar et al., 2021), and group the classes belonging to the same superclass into a vocabulary. There are 20 vocabularies in CIFAR100, each with 5 classes. For ImageNet, we utilize two superclass-class structures (Santurkar et al., 2021): Entity13 and Living17. The former has 13 vocabularies, each

---

[1]Since $\mathcal{V}^{(T)}$ is bound with $\mathcal{D}^{(T)}$ in Eq. 2, target vocabulary expansion implies expanding $\mathcal{D}^{(T)}$ (including input images and their labels) at the same time, which we omit for brevity.

Table 1: Extensibility and stability of CLIP-like models on CIFAR100 and ImageNet datasets. $\Delta$ refers to the decline of Acc-E/Acc-S (%) compared to Acc-C (%). PE denotes Prompt Ensemble. CoOp requires access to the training data in downstream datasets, and is prompt-tuned on all classes with 16 shots, which can be viewed as the upper bound of other zero-shot models.

| Model | CIFAR100 | | | | | ImageNet (Entity13) | | | | | ImageNet (Living17) | | | | |
| | | Extensibility | | Stability | | | Extensibility | | Stability | | | Extensibility | | Stability | |
| | Acc-C | Acc-E | $\Delta$ | Acc-S | $\Delta$ | Acc-C | Acc-E | $\Delta$ | Acc-S | $\Delta$ | Acc-C | Acc-E | $\Delta$ | Acc-S | $\Delta$ |
|---|---|---|---|---|---|---|---|---|---|---|---|---|---|---|---|
| CLIP (RN101) | 68.3 | 55.4 | -12.9 | 54.9 | -13.4 | 80.4 | 77.4 | -3.0 | 77.3 | -3.1 | 77.6 | 74.5 | -3.1 | 74.4 | -3.2 |
| CLIP (ViT-B/32) | 78.0 | 69.6 | -8.4 | 68.9 | -9.1 | 80.8 | 78.0 | -2.8 | 77.8 | -3.0 | 78.0 | 74.4 | -3.6 | 75.0 | -3.0 |
| CLIP (ViT-B/16) | **79.7** | **72.6** | **-7.1** | **72.0** | **-7.7** | 83.5 | 81.1 | -2.4 | 81.0 | -2.5 | 79.5 | 77.9 | -1.6 | 77.6 | -1.9 |
| SLIP (ViT-B/16) | 63.9 | 51.1 | -12.8 | 50.4 | -13.5 | 65.7 | 62.3 | -3.4 | 62.0 | -3.7 | 65.7 | 62.6 | -3.1 | 62.5 | -3.2 |
| DeCLIP (ViT-B/32) | **78.7** | **70.8** | **-7.9** | **70.4** | **-8.3** | 81.9 | 79.2 | -2.7 | 79.1 | -2.8 | 82.1 | 80.2 | -1.9 | 80.0 | -2.1 |
| PE (ViT-B/32) | 78.3 | 70.3 | -8.0 | 69.9 | -8.4 | 81.9 | 79.4 | -2.5 | 79.2 | -2.7 | 78.7 | 76.0 | -2.7 | 75.8 | -2.9 |
| PE (ViT-B/16) | **79.6** | **72.6** | **-7.0** | **72.0** | **-7.6** | 85.3 | 83.2 | -2.1 | 83.1 | -2.2 | 79.6 | 78.2 | -1.4 | 78.0 | -1.6 |
| CoOp (ViT-B/16) | 83.6 | 76.9 | -6.7 | 76.7 | -6.9 | 87.5 | 85.3 | -2.2 | 85.5 | -2.0 | 82.7 | 82.6 | -0.1 | 81.3 | -1.4 |

with 20 classes, while the latter has 17 vocabularies, each with 4 classes. [2] Tables in the Appendix A list all the vocabularies in the two datasets. For each dataset, we calculate Acc-C, the averaged classification accuracy across all single vocabularies, based on Eq. 2:

$$\text{Acc-C} = \frac{1}{N} \sum_{i=1}^{N} \text{Acc}\left(\mathcal{V}_i^{(T)}\right). \tag{4}$$

It represents the original model performance on *closed* vocabularies. To calculate the expectation in Acc-E, we sample $100 \times N$ permutations for $N$ vocabularies and take the average.

**Results** As shown in Table 1, all models exhibit a clear drop in performance as the openness of tasks increases. For example, on CIFAR100, compared with the accuracy on closed vocabulary (Acc-C), the accuracy after the vocabulary expansion (Acc-E) of CLIP (RN101) sharply decreased by 12.9%. The performance on the data splits in ImageNet is relatively better, with an averaged decline of 2.7%. Appendix B provides results of expansion at the dataset level where the expanded vocabularies are from five other datasets. The performance of CLIP-like models drops even more dramatically by an average of 15.3% on generic dataset expansion. These results demonstrate that **the openness of CLIP-like models is overestimated under the vanilla evaluation mechanism**. Besides, there are some interesting findings: **(1)** From the perspective of pre-training, introducing a stronger vision backbone (ViT (Dosovitskiy et al., 2021) v.s. ResNet (He et al., 2016)), widespread supervision (DeCLIP (Li et al., 2021b) v.s. CLIP), and more pre-training data (CLIP v.s. SLIP (Mu et al., 2021)) can improve the extensibility of models on open tasks. **(2)** During inference, the performance of CLIP can be boosted by ensembling different prompts (Radford et al., 2021). **(3)** CoOp (Zhou et al., 2021) that conducts prompt tuning on all classes of CIFAR100 and ImageNet yields the most extensible results. However, the prompt tuning method utilizes the predefined category information and training data in the target dataset, which cannot be transferred to real-life open tasks.

## 2.3 STABILITY DURING VOCABULARY EXPANSION

As the vocabulary expansion introduces new classes incrementally, some input images belonging to the previous vocabulary will be incorrectly predicted as new classes, which leads to an accuracy drop and the poor extensibility. To analyze the prediction stability of CLIP during vocabulary expansion, we introduce the *non-target classes*. They do NOT correspond to any input images, only serving as distractors for the target classes. Based on it, we define conditional classification accuracy as follows:

$$\text{Acc}\left(\mathcal{V}^{(T)} \middle| \mathcal{V}^{(T)} \cup \mathcal{V}^{(NT)}\right) = \frac{1}{|\mathcal{D}^{(T)}|} \sum_{(\mathbf{x},y)\in\mathcal{D}^{(T)}} \mathbb{I}\left(f\left(\mathbf{x}, \mathcal{V}^{(T)} \cup \mathcal{V}^{(NT)}\right) = y\right), \tag{5}$$

where $\mathcal{V}^{(NT)}$ is the *non-target vocabulary*, i.e., the vocabulary of non-target classes. The left panel of Figure 1 gives an illustration of the conditional accuracy. In Eq. 5, the categories of the input images

---

[2]Hence, the $N$ in Def. 2.1 for CIFAR100, ImageNet (Entity13) and (Living17) is 20, 13 and 17, respectively.

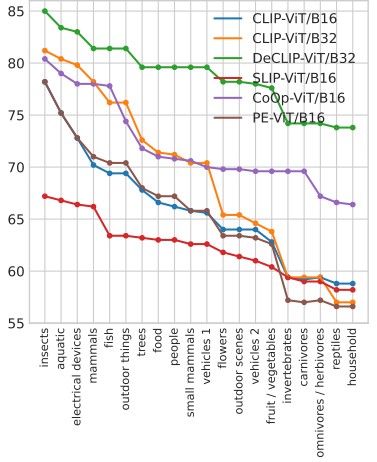 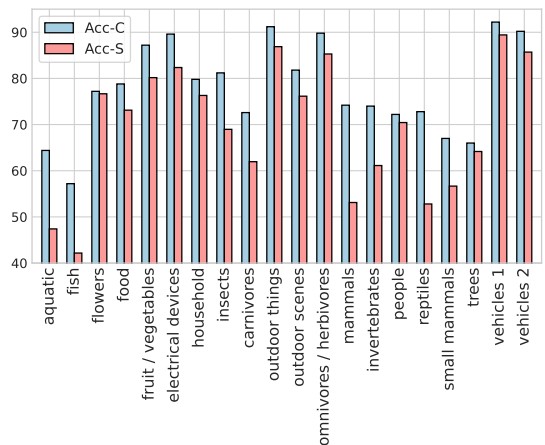

(a) Acc-S drops as non-target vocabulary extends (*Insects* as target vocabulary).

(b) Difference between Acc-C and Acc-S of CLIP (ViT-B/32) on different groups.

Figure 2: Acc-C and Acc-S (%) of CLIP and its variants on CIFAR100. The horizontal axis represents the extended non-target vocabularies in order. PE refers to Prompt Ensemble.

are limited to the target vocabulary ($(\mathbf{x}, y) \in \mathcal{D}^{(T)}$), but CLIP is asked to distinguish all categories from a broader vocabulary $\mathcal{V}^{(T)} \cup \mathcal{V}^{(NT)}$. In other words, compared to the traditional closed-set classification, CLIP is supposed to reject all the negative categories from $\mathcal{V}^{(NT)}$. The model is required to distinguish visual concepts stably and robustly, rather than making wrong predictions in the presence of other distractors. Based on Eq. 5, we define the stability of CLIP in the open task:

**Definition 2.2** (Stability). Given a target vocabulary $\mathcal{V}^{(T)}$ and $M$ non-target vocabularies $\{\mathcal{V}_1^{(NT)}, \cdots, \mathcal{V}_M^{(NT)}\}$, we denote $\mathcal{S}_M$ as their full permutation and $\mathcal{V}_{s_i}^{(NT)}$ as the $i^{(th)}$ vocabulary in a permutation $s \in \mathcal{S}_M$. We design the **local stability** to measure the averaged classification accuracy of CLIP on the given target vocabulary when non-target vocabularies are extended incrementally:

$$\text{Acc-}\tilde{\text{S}}\left(\mathcal{V}^{(T)}, \mathcal{V}^{(NT)}\right) = \mathop{\mathbb{E}}_{s \in \mathcal{S}_M} \frac{1}{M} \sum_{i=1}^{M} \text{Acc}\left(\mathcal{V}^{(T)} \middle| \mathcal{V}^{(T)} \cup \left(\mathcal{V}_{s_1}^{(NT)} \cup \cdots \cup \mathcal{V}_{s_i}^{(NT)}\right)\right). \quad (6)$$

As Eq. 6 only reflects the local stability with respect to a single target vocabulary, we further design the **general stability** as an average of local stability over a set of target vocabularies to reduce the bias from data distribution and vocabulary sampling. Specifically, given $N$ vocabularies $\{\mathcal{V}_1, \cdots, \mathcal{V}_N\}$, we regard each vocabulary $\mathcal{V}_i$ as the target vocabulary $\mathcal{V}^{(T)}$ and the rest $\mathcal{V}_{\neq i}$ as the non-target vocabularies $\mathcal{V}^{(NT)}$, and then formulate the general stability as:

$$\text{Acc-S} = \frac{1}{N} \sum_{i=1}^{N} \text{Acc-}\tilde{\text{S}}\left(\mathcal{V}_i, \mathcal{V}_{\neq i}\right). \quad (7)$$

**Experimental settings and results** The models and datasets adopted for evaluation are consistent with that in § 2.2. For the calculation of stability, take CIFAR100 with $N = 20$ vocabularies as an example, we treat each vocabulary as the target vocabulary and the rest are treated as the non-target vocabularies for Eq. 7. To calculate the expectation in Eq. 6, we sample 100 permutations for $M = 19$ non-target vocabularies and report the averaged scores.

Table 1 demonstrates the stability of CLIP-like models. On CIFAR100, the Acc-S of CLIP (RN101) decreased by $13.4\%$. Figure 2a shows Acc-S on CIFAR100 during non-target vocabulary expansion. Given a closed $\mathcal{V}^{(T)} = $ *Insects*, CLIP (ViT-B/32) achieves an accuracy of $81.2\%$. However, when the remaining 19 non-target vocabularies are incorporated, the accuracy sharply drops to $57.0\%$. The decrease of Acc-S brought by each introduction of non-target vocabulary indicates that more images from *Insects* are incorrectly classified into the new vocabulary. Figure 2b demonstrates the difference between Acc-C and Acc-S for each target vocabulary. When $\mathcal{V}^{(T)} = $ *Medium-sized Mammals*, CLIP

is most easily interfered with by the non-target vocabularies, with a $21.08\%$ performance drop. It suggests that **the unstable predictions lead to the poor extensibility of CLIP when new categories are introduced**. Besides, we notice that CLIP performs stably on groups like *Flowers*, where its Acc-S only declines by $0.53\%$ compared to Acc-C. The different behaviors of different groups indicates that **the stability is also influenced by the inherent property of the image categories**.

Furthermore, In order to explore the lower bound of the stability, we define the *adversarial non-target vocabulary* $\mathcal{V}^{(ANT)}$ as the non-target vocabulary that reduces Acc-S the most. Specifically, we maliciously introduce only three new classes and find that the performance of CLIP drops dramatically, e.g., $52.7\%$ accuracy drop on CIFAR10. It reveals the vulnerability of CLIP and its poor semantic modeling on those objects with higher abstraction levels. Please refer to Appendix D for details.

## 3 DISSECTING AND IMPROVING THE EXTENSIBILITY OF CLIP

Our experimental results in § 2 expose the unsatisfying performance of CLIP on open tasks. In this section, we dive into the representation space of CLIP to find the key to understanding and improving its extensibility. We first point out that the small margin between positive and negative class descriptions leads to the prediction shift when competing text features appear, which thus limits the stability of CLIP (§ 3.1). Further, we investigate the representation space of CLIP-like models via two metrics of inter-modal alignment and intra-modal uniformity. The results show that enforcing the distinguishability of text features enlarges the margin and makes models scale more stably (§ 3.2). In response, we propose a *non-parametric* method named Retrieval-Enhanced Prompt Engineering (REPE), which boosts the performance of CLIP even without fine-tuning (§ 3.3).

### 3.1 SMALL MARGIN LIMITS THE STABILITY OF CLIP

Since CLIP formalizes the visual recognition as an image-text matching task (Eq. 1), each text feature of the class description corresponds to the class vector in traditional classifiers, and the image-text similarity scores are thus analogous to the logits in classification. Ideally, no matter how the vocabulary expands, for an image, the similarity of the positive pair, i.e., the image with the text specifying the ground-truth class, should be higher than those of the negative pairs, i.e., the image with the texts specifying other classes, to ensure the correct prediction of CLIP on open tasks. In other words, the *margin* (Jiang et al., 2019) between positive and the largest negative similarity is a direct contributing factor to the stability.

Unfortunately, the similarity and margin distribution of CLIP does not meet our expectations. Figure 3 illustrates the averaged cosine similarity of CLIP (ViT-B/32) on 15 classes of CIFAR100. Similarity over the intact dataset is in Appendix E. The diagonal elements represent the similarity of the positive image-text pairs, while the others represent that of the negative ones. As shown in Figure 4, the similarity histogram of positive and negative pairs has a large overlap. Its margin in Figure 5 is clustered around zero, leaving the prediction of models at risk of being reversed to the new non-target classes. For example, as the vocabulary extends from the red box to the green box (diagonal) or the yellow box (horizontal) in Figure 3, more deceptive classes (circle) with negative margin are added, which leads to the prediction shift. Particularly, the classes belonging to the same vocabulary [3] have higher similarity and smaller margin, which is more likely to be confused with each other.

### 3.2 INTER-MODAL ALIGNMENT AND INTRA-MODAL UNIFORMITY GROUND THE MARGIN

The results in § 3.1 raise a natural question: what vision-and-language feature distribution can maintain a large margin between different classes so that the model can scale stably in the open world? Here we present two properties of the ideal feature space: First, the text feature of a class name is supposed to stay close to the features of the images it describes, promoting the similarity of positive pairs. Second, intra-modal features, especially the textual features should be uniformly distributed to preserve maximal information and make the descriptions of competing categories more distinguishable. Accordingly, we define **inter-modal alignment** and **intra-modal uniformity**, two metrics to measure the quality of representations in contrastive learning (Wang & Isola, 2020) for the vision-and-language domain. Inter-modal alignment calculates the expected distance between

---

[3]Every 5 adjacent classes in Figure 3 constitute a vocabulary (superclass), see Table 3 in Appendix A

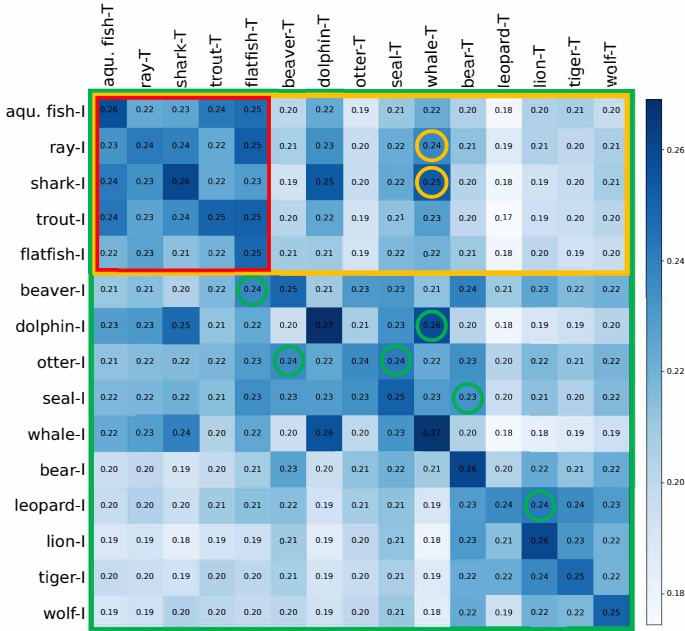

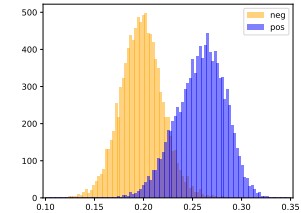

Figure 4: Cosine similarity histogram of positive (pos) and negative (neg) image-text pairs with large overlap.

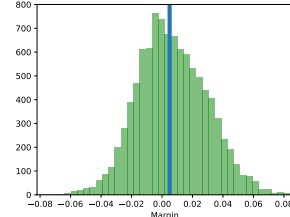

Figure 3: Averaged cosine similarity between image (-I) and text (-T) features of CLIP (ViT-B/32) on CIFAR100. The expansion from the red box to the green box (diagonal) and the yellow box (horizontal) refer to the calculation of extensibility and stability, respectively. The circle represents that more than 15 wrong predictions have arisen after adding this class.

Figure 5: Margin distribution of similarity scores, which are centered around zero with a median value of .005 (the blue vertical line). It indicates that the predictions can be easily inverted with competing classes appearing.

features of positive image-text pairs $p_{\text{pos}}$ :

$$\ell_{\text{align}} \triangleq \mathop{\mathbb{E}}_{(\mathbf{x},\mathbf{t})\sim p_{\text{pos}}} \|f_I(\mathbf{x}) - f_T(\mathbf{t})\|^2 , \tag{8}$$

while intra-modal uniformity measures how well the image or text features are uniformly distributed:

$$\begin{aligned}
\ell_{\text{uniform}} &\triangleq \ell_{\text{uniform-I}} + \ell_{\text{uniform-T}} \\
&\triangleq \log \mathop{\mathbb{E}}_{\mathbf{x}_i,\mathbf{x}_j \overset{i.i.d}{\sim} p_{\text{data-I}}} e^{-2\|f_I(\mathbf{x}_i)-f_I(\mathbf{x}_j)\|^2} + \log \mathop{\mathbb{E}}_{\mathbf{t}_i,\mathbf{t}_j \overset{i.i.d}{\sim} p_{\text{data-T}}} e^{-2\|f_T(\mathbf{t}_i)-f_T(\mathbf{t}_j)\|^2} ,
\end{aligned} \tag{9}$$

where $p_{\text{data-I}}$ and $p_{\text{data-T}}$ denotes the image and text data distribution, respectively. Figure 6 and Table 7 in Appendix F provide quantified loss of alignment and uniformity. CLIP with only cross-modal contrastive learning results in poor intra-modal uniformity ($\ell_{\text{uniform}} > -2.0$), especially on the text side. Introducing intra-modal contrastive learning like SLIP and DeCLIP in pre-training can force both image and text features separate better from classes, reducing $\ell_{\text{uniform}}$ to below $-4.5$. As for the prompt tuning method (CoOp), it achieves better $\ell_{\text{align}}$ of $1.4$ compared to CLIP ($1.5$) and the lowest $\ell_{\text{uniform-T}}$ of $-3.2$. According to the visualization via Multidimensional Scaling (MDS) (Borg & Groenen, 1997) in Figure 7, the optimization trajectory of prompts is towards the cluster center of the corresponding image features, while dispersing the position of the prompt features, which improves both text uniformity and inter-modal alignment, achieving the best extensibility.

### 3.3 METHODOLOGY: RETRIEVAL-ENHANCED PROMPT ENGINEERING (REPE)

In light of the previous investigations, we propose a simple but effective method named Retrieval-enhanced Prompt Engineering (REPE) to enforce the distinguishability of text features and the semantic alignment (Cao et al., 2020; Ren et al., 2021). Recall that the context for each class name is the same in vanilla CLIP-like models (e.g., "a photo of a [CLASS]"), making it difficult to discriminate between distinct visual categories because the semantics of each cannot be holistic represented Zhou et al. (2022). To remedy this, we propose to customize each class description with

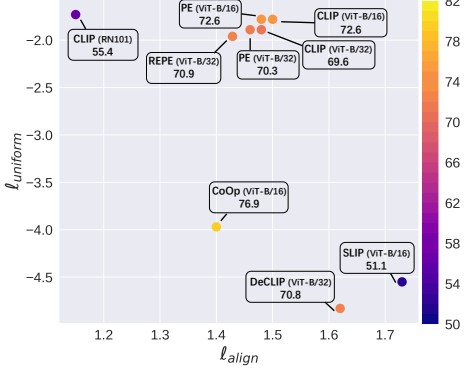

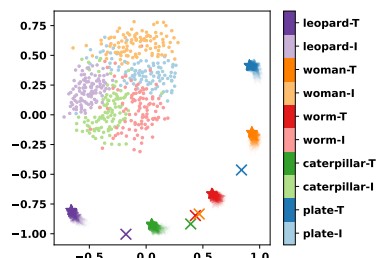

Figure 7: Representation visualization of CLIP and CoOp (ViT-B/16). The five classes with different colors are from CIFAR100. ● refers to image features (-I), while × and ⋆ refers to text features (-T) of CLIP and CoOp, respectively. The color of ⋆ from transparent to opaque indicates the optimization trajectory during the CoOp prompt-tuning process.

Figure 6: $\ell_{\text{align}}$ and $\ell_{\text{uniform}}$ of CLIP-like models. For both two metrics, lower numbers are better. The color of points and numbers denote the extensibility performance (Acc-E) on CIFAR100 (higher is better).

Table 2: Extensibility and stability of our REPE method on CIFAR100 and ImageNet datasets.

| Model | CIFAR100 | | | ImageNet (Entity13) | | | ImageNet (Living17) | | |
|---|---|---|---|---|---|---|---|---|---|
| | Acc-C | Acc-E | Acc-S | Acc-C | Acc-E | Acc-S | Acc-C | Acc-E | Acc-S |
| CLIP (RN101) | 68.3 | 55.4 | 54.9 | 80.4 | 77.4 | 77.3 | 77.6 | 74.5 | 74.4 |
| REPE (RN101) | **68.4** (+0.1) | **55.5** (+0.1) | **55.2** (+0.3) | **81.7** (+1.3) | **79.2** (+1.8) | **79.0** (+1.7) | **77.8** (+0.2) | **75.3** (+0.8) | **75.2** (+0.8) |
| CLIP (ViT-B/32) | 78.0 | 69.6 | 68.9 | 80.8 | 78.0 | 77.8 | 78.0 | 74.4 | 75.0 |
| REPE (ViT-B/32) | **78.5** (+0.5) | **70.9** (+1.3) | **70.6** (+1.7) | **82.3** (+1.5) | **79.8** (+1.8) | **79.6** (+1.8) | **79.0** (+1.0) | **76.4** (+2.0) | **76.2** (+1.2) |
| CLIP (ViT-B/16) | 79.7 | 72.6 | 72.0 | 83.5 | 81.1 | 81.0 | 79.5 | 77.9 | 77.6 |
| REPE (ViT-B/16) | **79.8** (+0.1) | **72.9** (+0.3) | **72.6** (+0.6) | **85.4** (+1.9) | **83.3** (+2.2) | **83.2** (+2.2) | **79.9** (+0.4) | **78.4** (+0.5) | **78.2** (+0.6) |

**diverse captions retrieved from the pre-training corpus as a prompt ensemble.** Specifically, for each class description based on the original prompt, we utilize CLIP to recall the most similar images from the pre-training dataset via image-text similarity then obtain their corresponding captions. The retrieved captions without class name appearance are filtered out, finally resulting in $K$ captions. Such a workflow leverages both visual semantics and class name, achieving better performance. Appendix G shows the instances of the retrieved captions. They provide the context in which the class name is located and thus have richer semantics. After that, we encode the retrieved captions and conduct a mean pooling operation among them. The final text representation is

$$f_T^{\text{REPE}}(\mathbf{t}_i) = (1 - \lambda)f_T(\mathbf{t}_i) + \lambda \frac{1}{K} \sum_{j}^{K} f_T(\mathbf{rt}_{ij}), \tag{10}$$

where $\mathbf{rt}_{ij}$ is the $j^{(th)}$ retrieved caption for class $i$, $\lambda$ is a weighting factor and is adjusted on the validation set. The ensemble text representation $f_T^{\text{REPE}}(\mathbf{t}_i)$ is then adopted as the class anchor for conducting the image classification. The text representation for the target class is thus shifted towards the representative captions in the pre-training dataset, which alleviates the semantic inconsistency between pre-training and inference.

**Experiments** We retrieve the images and captions from CC12M (Changpinyo et al., 2021), a subset of the pre-training dataset of CLIP. The images and captions are pre-encoded within **an hour** using a single RTX TITAN GPU, then we build their indices for KNN search with FAISS framework (Johnson et al., 2019), which also takes about **an hour**. Once the indices are built, we can efficiently search over the dataset according to the query image in less than **5 ms**, which is applicable for query-intensive scenarios. Appendix H provides the detailed process and computational overhead of the retrieval. Table 2 shows the results of retrieval-enhanced prompt engineering. The hyper-parameter $K$ is 100 and $\lambda$ is 0.25. We find that on all the three datasets, REPE consistently improves the extensibility and stability of CLIP by an average of **1.2%**. We further evaluate the loss of text uniformity and inter-modal alignment for probing the quality of the enhanced representations. As shown in Figure 6, the former is effectively reduced from $-0.8$ to $-1.0$ and the latter is reduced from 1.5 to 1.4, verifying

our proposal can improve the class anchor for better extensibility and stability. Additionally, REPE increases the median value of the margin distribution from 0.005 to 0.01 and pushes the overall distribution towards the positive side compared to vanilla CLIP (Figure 12 in Appendix E). It indicates that REPE widens the gap between positive and negative class features, making it more difficult to invert predictions with competing classes. In conclusion, all of these findings support REPE's efficacy in addressing the openness issue.

It is worth noting that compared to the method that requires computation-intensive pre-training procedure (DeCLIP and SLIP), and the prompt-tuning approach (CoOp) demands access to the downstream target dataset, our REPE is a lightweight framework for the zero-shot inference stage without fine-tuning. Besides, since REPE is model-agnostic and orthogonal to parameter-tuning methods, it can be also combined with fine-tuning methods like adapter-tuning (Gao et al., 2021), to achieve further performance boost by an average of $0.6$ on CIFAR100 and ImageNet, which demonstrates the adaptability and superiority of our method. Please refer to Appendix I for details.

## 4 Related work

**Contrastive language-image pre-training and open-vocabulary learning**    CLIP (Radford et al., 2021) enables learning transferable visual models from natural language supervision and makes visual recognition generalize in the wild (Zareian et al., 2021; Gu et al., 2022; Ghiasi et al., 2021). It is pre-trained on web-scale collections of image-text pairs, learning tremendous visual concepts described by natural language with contrastive learning. During inference, it devises a prompt like "`a photo of a [CLASS]`", where the class token is a placeholder for any potential class name from a vocabulary, and the class description with the highest similarity to the input image is predicted as the target class. Another line of recent studies (Li et al., 2021a; Wang et al., 2022; Yu et al., 2022; Alayrac et al., 2022) adopts seq2seq generation instead of contrastive discrimination framework to achieve open-vocabulary recognition. We leave the investigation of their extensibility for future work.

**Open set and open-world visual recognition**    Open Set Recognition (OSR) (Scheirer et al., 2013; Geng et al., 2021) requires classifiers to identify images that have not been introduced during training as "unknown". The task can be formalized as one-vs-reset classification (Scheirer et al., 2013) or multi-class classification (Jain et al., 2014; Scheirer et al., 2014). Furthermore, Open World Recognition (OWR) (Bendale & Boult, 2015) raises higher demands that the model must incrementally learn and extend the multi-class classifier as the unknowns are labeled for new class learning. Contrary to the above research, the CLIP-based Open-vocabulary Recognition (OVR) is unsupervised. We focus on the model performance on zero-shot inference, without training on the target dataset. Appendix J provides a more detailed comparison of OSR, OWR, and OVR.

## 5 Limitations and Future Work

To facilitate future research, we analyze the difficulties and possible solutions in this new area. **(1)** As we present extensive empirical results and address the weakness of CLIP on vocabulary expansion, its theoretical risk on open tasks is urged to be investigated. **(2)** The current evaluation protocol is an approximation of the real open world. An evolving benchmark could facilitate future research. **(3)** For various visual categories, their degree of abstraction, the ease of describing them in natural language, and their density in the data distribution can also influence the extensibility and stability of models, which are worth studying. **(4)** Within the CLIP & prompting framework, our REPE can be easily adopted in various downstream visual tasks like open-vocabulary object detection (Zareian et al., 2021), semantic segmentation (Ghiasi et al., 2021), etc. We leave it for future work.

## 6 Conclusion

In this paper, we evaluate the extensibility of CLIP-like models for open-vocabulary visual recognition. Our extensive investigation shows that the performance deteriorates seriously as the vocabulary expands, which results from the indistinguishable text features among competing classes. To remedy this, we propose REPE to enhance the textual representations with class-relevant captions retrieved from the pre-training corpus, which boosts the extensibility and stability even without fine-tuning.

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

## A  SUPERCLASS-CLASS HIERARCHY FOR VOCABULARY CONSTRUCTION

To construct the vocabulary in § 2, we leverage the underlying superclass-class hierarchical structure of CIFAR100 (Krizhevsky & Hinton, 2009) and ImageNet (Deng et al., 2009), and group the classes belonging to the same superclass into a vocabulary. Table 3 lists the vocabularies in CIFAR100, which are specified by (Krizhevsky & Hinton, 2009). There are 20 vocabularies, each with 5 classes.

For ImageNet, we utilize two superclass-class structures, Entity13 and Living17, in (Santurkar et al., 2021). Table 4 and Table 5 show the vocabularies in ImageNet (Entity13) and ImageNet (Living17), respectively. The former has 13 vocabularies, each with 20 classes, while the latter has 17 vocabularies, each with 4 classes.

## B    DATASET-LEVEL EXTENSIBILITY

The evaluation protocol in § 2 estimates the extensibility and stability within a single task dataset, where the input images and classes during the vocabulary expansion come from the same data distribution. While the protocol is only an approximation of the real open world, current CLIP-like models have exhibited serious performance degradation. In this section, we take a step further toward the real open recognition, by conducting vocabulary expansion setting on the dataset level, where the expanded vocabularies are from different datasets. In this way, the relationship between vocabularies is more uncertain and thus can be viewed as a rigorous stress test for the CLIP-like models. Specifically, we group all categories in a dataset into one vocabulary. Afterward, the inputs and classes of the entire new dataset are introduced at each expansion. Classes in the new vocabulary will be removed if they already exist in the previous vocabularies.

The experiments are conducted with datasets for generic objects, including CIFAR10 (Krizhevsky & Hinton, 2009), CIFAR100 (Krizhevsky & Hinton, 2009), Caltech101 (Fei-Fei et al., 2004), SUN397 (Xiao et al., 2010) and ImageNet (Deng et al., 2009), and specialized datasets focusing on fine-grained categories, including Flowers102 (Nilsback & Zisserman, 2008), OxfordPets (Parkhi et al., 2012) and StanfordCars (Krause et al., 2013). Without loss of the generality, we merge 3 datasets and evaluate the following dataset compositions:

(1)  CIFAR100-Caltech101-SUN397
(2)  CIFAR10-CIFAR100-ImageNet
(3)  Flowers102-OxfordPets-StanfordCars

Composition (1) and (2) probe the performance when all the expanded datasets are generic thus the classes in different datasets are semantics correlated, while composition (3) targets scenarios where the coming datasets have little correlation with previous ones. To eliminate the effect of vocabulary expansion order, we report the average performance of all $A_3^3 = 6$ possible trials for each composition.

Table 6 demonstrates the result of the dataset-level expansion. **First**, the performance of CLIP-like models on generic dataset expansion drops dramatically. For example, the accuracy (Acc-E) of CLIP (RN101) decreases by an averaged absolute point of 14.2 on the *CIFAR100-Caltech101-SUN397* composition during expansion, and 14.5 on the *CIFAR10-CIFAR100-ImageNet* composition, respectively. Due to the existence of subclass-superclass relationship for some classes in different generic datasets, e.g., *cat* in CIFAR10 and *tiger cat* in ImageNet, CLIP is extremely unstable on such expansion across generic datasets. For example, the Acc-S of CLIP (RN101) on the *CIFAR10-CIFAR100-ImageNet* composition is 28.2% lower than Acc-C, indicating the models are prone to be confused about the subclass-superclass relationship. **Meanwhile**, the CLIP-like models exhibit much better extensibility and stability on the dataset-level expansion across specialized datasets, e.g., the *Flowers102-OxfordPets-StanfordCar* composition. The vocabularies of this composition are intrinsically disjoint in semantics, so the model can be stably extended. **In summary**, our investigations on the dataset level expansions along with the task level in the paper show the current CLIP-like models fail to meet the expectation of conducting real open vocabulary recognition, and we hope our studies can motivate future studies in this direction.

## C    INCREMENTAL ACC-E AND ACC-S ON CIFAR100

We record the Acc-E (Eq. 3) and Acc-S (Eq. 6) after each vocabulary expansion on CIFAR100 to investigate the effect of increased task openness on CLIP-like models.

Figure 8 shows the Acc-E for 20 trials as new vocabularies are merged incrementally. The falling lines indicate that the model is either performing poorly on the new input images, or that some images that were correctly identified before are misclassified after introducing the new classes.

Table 3: Superclass-class hierarchy in CIFAR100. Each superclass corresponds to a vocabulary, and each vocabulary has 5 classes. There are 20 kinds of vocabulary in total, specified by Krizhevsky & Hinton (2009).

| Vocabulary (Superclass) | Classes |
|---|---|
| aquatic | mammals beaver, dolphin, otter, seal, whale |
| fish | aquarium fish, flatfish, ray, shark, trout |
| flowers | orchids, poppies, roses, sunflowers, tulips |
| food | containers bottles, bowls, cans, cups, plates |
| fruit and vegetables | apples, mushrooms, oranges, pears, sweet peppers |
| household electrical devices | clock, computer keyboard, lamp, telephone, television |
| household | furniture bed, chair, couch, table, wardrobe |
| insects | bee, beetle, butterfly, caterpillar, cockroach |
| large carnivores | bear, leopard, lion, tiger, wolf |
| large man-made outdoor things | bridge, castle, house, road, skyscraper |
| large natural outdoor scenes | cloud, forest, mountain, plain, sea |
| large omnivores and herbivores | camel, cattle, chimpanzee, elephant, kangaroo |
| medium-sized mammals | fox, porcupine, possum, raccoon, skunk |
| non-insect invertebrates | crab, lobster, snail, spider, worm |
| people | baby, boy, girl, man, woman |
| reptiles | crocodile, dinosaur, lizard, snake, turtle |
| small mammals | hamster, mouse, rabbit, shrew, squirrel |
| trees | maple, oak, palm, pine, willow |
| vehicles 1 | bicycle, bus, motorcycle, pickup truck, train |
| vehicles 2 | lawn-mower, rocket, streetcar, tank, tractor |

Figure 9 shows Acc-S of CLIP-like models during non-target vocabulary expansion. Each sub-figure represents the situation when one vocabulary is selected as the target vocabulary. As the remaining 19 non-target vocabularies are incorporated and the model is required to recognize the 5 target classes from 100 potential classes, the accuracy drops sharply. The decrease of Acc-S brought by each introduction of non-target vocabulary indicates that more images from the target vocabulary are incorrectly classified into the new non-target vocabulary by models.

## D  ADVERSARIAL NON-TARGET VOCABULARY

In order to explore the lower bound of the stability of CLIP, we define the *adversarial non-target vocabulary* $\mathcal{V}^{(ANT)}$ as the non-target vocabulary that reduces Acc-S the most:

$$\mathcal{V}^{(ANT)} = \min_{\mathcal{V}^{(NT)}} \text{Acc}\left(\mathcal{V}^{(T)}\Big|\mathcal{V}^{(T)} \cup \mathcal{V}^{(NT)}\right). \tag{11}$$

To build $\mathcal{V}^{(ANT)}$, we refer to the method of adversarial examples generation in the Natural Language Processing field (Ren et al., 2019) to traverse the words in a large vocabulary, e.g., the vocabulary of nouns in WordNet (Fellbaum, 2000), which are regarded as non-target classes in order to calculate Acc-S, and then take the most confusing words to form the adversarial non-target vocabulary.

We constrain the size of $\mathcal{V}^{(ANT)}$ to 3. Results in Figure 10 illustrate the performance with nouns in WordNet and class names in ImageNet as the candidate vocabulary, respectively. First, we observe a clear performance degradation on both datasets under adversarial attack, e.g., adding *bitmap*, *automobile insurance* and *equidae* leads to an absolute 52.7% accuracy drop on CIFAR10. Besides, we find that the selected adversarial words are much less concrete than common visual concepts like *Flower*, indicating the potential reason behind is the poor semantic modeling of CLIP on those objects with higher abstraction levels. This investigation reveals that **CLIP is vulnerable when facing malicious non-target vocabulary**, and we hope future work may pay more attention to the robustness of CLIP under open recognition tasks.

Table 4: Superclass-class hierarchy in ImageNet (Entity13). Each superclass corresponds to a vocabulary, and each vocabulary has 20 classes. There are 13 kinds of vocabulary in total, specified by BREEDS Santurkar et al. (2021).

| Vocabulary (Superclass) | Classes |
|---|---|
| garment | trench coat, abaya, gown, poncho, military uniform, jersey, cloak, bikini, miniskirt, swimming trunks, lab coat, brassiere, hoopskirt, cardigan, pajama, academic gown, apron, diaper, sweatshirt, sarong |
| bird | African grey, bee eater, coucal, American coot, indigo bunting, king penguin, spoonbill, limpkin, quail, kite, prairie chicken, red-breasted merganser, albatross, water ouzel, goose, oystercatcher, American egret, hen, lorikeet, ruffed grouse |
| reptile | Gila monster, agama, triceratops, African chameleon, thunder snake, Indian cobra, green snake, mud turtle, water snake, loggerhead, sidewinder, leatherback turtle, boa constrictor, garter snake, terrapin, box turtle, ringneck snake, rock python, American chameleon, green lizard |
| arthropod | rock crab, black and gold garden spider, tiger beetle, black widow, barn spider, leafhopper, ground beetle, fiddler crab, bee, walking stick, cabbage butterfly, admiral, lacewing, trilobite, sulphur butterfly, cicada, garden spider, leaf beetle, long-horned beetle, fly |
| mammal | Siamese cat, ibex, tiger, hippopotamus, Norwegian elkhound, dugong, colobus, Samoyed, Persian cat, Irish wolfhound, English setter, llama, lesser panda, armadillo, indri, giant schnauzer, pug, Doberman, American Staffordshire terrier, beagle |
| accessory | bib, feather boa, stole, plastic bag, bathing cap, cowboy boot, necklace, crash helmet, gasmask, maillot, hair slide, umbrella, pickelhaube, mit- ten, sombrero, shower cap, sock, running shoe, mortarboard, handkerchief |
| craft | catamaran, speedboat, fireboat, yawl, airliner, container ship, liner, trimaran, space shuttle, aircraft carrier, schooner, gondola, canoe, wreck, warplane, balloon, submarine, pirate, lifeboat, airship |
| equipment | volleyball, notebook, basketball, hand-held computer, tripod, projector, barbell, monitor, croquet ball, balance beam, cassette player, snorkel, horizontal bar, soccer ball, racket, baseball, joystick, microphone, tape player, reflex camera |
| furniture | wardrobe, toilet seat, file, mosquito net, four-poster, bassinet, chiffonier, folding chair, fire screen, shoji, studio couch, throne, crib, rocking chair, dining table, park bench, chest, window screen, medicine chest, barber chair |
| instrument | upright, padlock, lighter, steel drum, parking meter, cleaver, syringe, abacus, scale, corkscrew, maraca, saltshaker, magnetic compass, accordion, digital clock, screw, can opener, odometer, organ, screwdriver |
| man-made structure | castle, bell cote, fountain, planetarium, traffic light, breakwater, cliff dwelling, monastery, prison, water tower, suspension bridge, worm fence, turnstile, tile roof, beacon, street sign, maze, chain-link fence, bakery, drilling platform |
| wheeled vehicle | snowplow, trailer truck, racer, shopping cart, unicycle, motor scooter, passenger car, minibus, jeep, recreational vehicle, jinrikisha, golfcart, tow truck, ambulance, bullet train, fire engine, horse cart, streetcar, tank, Model T |
| produce | broccoli, corn, orange, cucumber, spaghetti squash, butternut squash, acorn squash, cauliflower, bell pepper, fig, pomegranate, mushroom, strawberry, lemon, head cabbage, Granny Smith, hip, ear, banana, artichoke |

Table 5: Superclass-class hierarchy in ImageNet (Living17). Each superclass corresponds to a vocabulary, and each vocabulary has 4 classes. There are 17 kinds of vocabulary in total, specified by BREEDS Santurkar et al. (2021).

| Vocabulary (Superclass) | Classes |
|---|---|
| salamander | eft, axolotl, common newt, spotted salamander |
| turtle | box turtle, leatherback turtle, loggerhead, mud turtle |
| lizard | whiptail, alligator lizard, African chameleon, banded gecko |
| snake | night snake, garter snake, sea snake, boa constrictor |
| spider | tarantula, black and gold garden spider, garden spider, wolf spider |
| grouse | ptarmigan, prairie chicken, ruffed grouse, black grouse |
| parrot | macaw, lorikeet, African grey, sulphur-crested cockatoo |
| crab | Dungeness crab, fiddler crab, rock crab, king crab |
| dog | bloodhound, Pekinese, Great Pyrenees, papillon |
| wolf | coyote, red wolf, white wolf, timber wolf |
| fox | grey fox, Arctic fox, red fox, kit fox |
| domestic cat | tiger cat, Egyptian cat, Persian cat, Siamese cat |
| bear | sloth bear, American black bear, ice bear, brown bear |
| beetle | dung beetle, rhinoceros beetle, ground beetle, long-horned beetle |
| butterfly | sulphur butterfly, admiral, cabbage butterfly, ringlet |
| ape | gibbon, orangutan, gorilla, chimpanzee |
| monkey | marmoset, titi, spider monkey, howler monkey |

Table 6: Extensibility and stability of CLIP and its variants during dataset-level vocabulary expansion. $\Delta$ refers to the decline of Acc-E/Acc-S (%) compared to Acc-C (%). PE denotes Prompt Ensemble.

| Model | CIFAR100-Caltech101-SUN397 | | | | | CIFAR10-CIFAR100-ImageNet | | | | | Flowers102-OxfordPets-StanfordCars | | | | |
|---|---|---|---|---|---|---|---|---|---|---|---|---|---|---|---|
| | Acc-C | Extensibility | | Stability | | Acc-C | Extensibility | | Stability | | Acc-C | Extensibility | | Stability | |
| | | Acc-E | $\Delta$ | Acc-S | $\Delta$ | | Acc-E | $\Delta$ | Acc-S | $\Delta$ | | Acc-E | $\Delta$ | Acc-S | $\Delta$ |
| CLIP (RN101) | 65.9 | 51.7 | -14.2 | 52.7 | -13.2 | 62.4 | 47.9 | **-14.5** | 34.2 | **-28.2** | 65.8 | 63.1 | -2.7 | 65.7 | -0.1 |
| CLIP (ViT-B/32) | 72.0 | 59.4 | **-12.6** | 61.2 | **-10.8** | 70.9 | 52.7 | -18.2 | 41.3 | -29.6 | 65.8 | 62.0 | -3.8 | 65.8 | **-0.0** |
| CLIP (ViT-B/16) | 74.6 | **60.6** | -14.0 | **61.7** | -12.9 | 74.7 | **56.6** | -18.0 | **43.3** | -31.4 | 72.3 | **69.6** | -2.7 | **72.3** | **-0.0** |
| SLIP (ViT-B/16) | 58.6 | 44.4 | -14.2 | 46.3 | -12.3 | 55.6 | 36.7 | -18.9 | 30.5 | **-25.1** | 35.0 | 26.0 | -9.0 | 35.0 | **-0.0** |
| DeCLIP (ViT-B/32) | 74.3 | 60.8 | **-13.5** | 63.3 | **-11.0** | 73.0 | 55.4 | -17.6 | 45.1 | -27.9 | 70.2 | 63.3 | -6.9 | 70.2 | **-0.0** |
| PE (ViT-B/32) | 71.8 | 59.9 | **-11.9** | 59.6 | -12.2 | 72.2 | 53.5 | **-18.7** | 41.6 | -30.6 | 65.7 | 62.0 | -3.7 | 65.7 | **-0.0** |
| PE (ViT-B/16) | **75.0** | 61.5 | -13.5 | **62.5** | -12.5 | **75.4** | 56.7 | **-18.7** | 41.3 | -34.1 | 72.5 | 70.0 | -2.5 | 72.5 | **-0.0** |

Table 7: Inter-modal alignment ($\ell_{\text{align}}$), text uniformity ($\ell_{\text{uniform-T}}$), image uniformity ($\ell_{\text{uniform-I}}$), Acc-C (Eq. 4), and Acc-E (Eq. 3) of CLIP-like models on CIFAR100. For the first three metrics, lower numbers are better. For the last two metrics, higher numbers are better.

| Model | Alignment & Uniformity | | | Accuracy | |
|---|---|---|---|---|---|
| | $\ell_{\text{align}}$ ($\downarrow$) | $\ell_{\text{uniform-T}}$ ($\downarrow$) | $\ell_{\text{uniform-I}}$ ($\downarrow$) | Acc-C ($\uparrow$) | Acc-E ($\uparrow$) |
| CLIP (RN101) | **1.15** | **-1.16** | -0.57 | 68.3 | 55.4 |
| CLIP (ViT-B/32) | 1.48 | -0.96 | **-0.93** | 78.0 | 69.6 |
| CLIP (ViT-B/16) | 1.50 | -0.97 | -0.81 | **79.7** | **72.6** |
| SLIP (ViT-B/16) | 1.73 | -2.86 | -1.69 | 63.9 | 51.1 |
| DeCLIP (ViT-B/32) | **1.62** | **-2.96** | **-1.87** | **78.7** | **70.8** |
| PE (ViT-B/32) | **1.46** | -0.96 | **-0.93** | 78.3 | 70.3 |
| PE (ViT-B/16) | 1.48 | **-0.97** | -0.81 | **79.6** | **72.6** |
| CoOp (ViT-B/16) | 1.40 | -3.16 | -0.81 | 83.6 | 76.9 |

## E    COSINE SIMILARITY OF IMAGE-TEXT FEATURES ON INTACT CIFAR100

Figure 11 illustrates the averaged cosine similarity between image and text features of CLIP (ViT-B/32) on all classes of CIFAR100. The elements on the diagonal represent the similarity of the positive image-text pairs, while the others represent that of the negative ones. Since every 5 adjacent classes in the figure constitute a vocabulary (superclass), [4] the classes belonging to the same vocabulary have higher similarity and smaller margin, which is more likely to be confused with each other.

Our Retrieval-enhanced Prompt Engineering (REPE) method alleviates this issue by enlarging the margin between positive and the largest negative similarity. As shown in Figure 12, the median value of REPE's margin distribution is .01 (the blue vertical line), which is larger than that of CLIP (ViT-B/32) with .005 (the red line). It indicates that the predictions of REPE are harder to be inverted with competing classes than the original CLIP, thus yielding better performance on the open tasks.

## F    QUANTIFIED LOSS OF INTER-MODAL ALIGNMENT AND INTRA-MODAL UNIFORMITY

Table 7 provides quantified loss of alignment and uniformity based on Eq. 8 and Eq. 9 defined in § 3.2. CLIP with only cross-modal contrastive learning results in poor intra-modal uniformity. On the vision side, compared with ResNet-101 (He et al., 2016), using a more powerful visual encoder such as ViT (Dosovitskiy et al., 2021) can reduce the loss of image uniformity from $-0.57$ to $-0.93$. Besides, the $\ell_{\text{uniform}}$ of SLIP (Mu et al., 2021) and DeCLIP (Li et al., 2021b) is much lower than CLIP, indicating their better intra-modal uniformity derived by intra-modal contrastive learning in pre-training, which enforces both image and text features separate better from classes. As for the prompt tuning (Zhou et al., 2021) method (CoOp), it achieves better $\ell_{\text{align}}$ compared to CLIP and the lowest $\ell_{\text{uniform-T}}$ of $-3.16$.

## G    CASE STUDY OF RETRIEVED CAPTION IN REPE

Table 8 shows some cases of the captions retrieved by our proposed REPE on CIFAR100. They share the same target of interest with the original prompt, i.e., "`a photo of a [CLASS]`", but provide the context in which the class name is located and thus have richer semantics. For example, given a class like *bridge*, the retrieved captions describe its possible properties (e.g., "golden", "wooded"), connections to other objects (e.g., "over a mountain river"), etc., yielding more expressive and distinguishable text features of the class. However, REPE also recalls some low-quality captions. For example, given the class *ray*, a large, flat sea fish with a long, narrow tail, in CIFAR100, the caption "Sun Rays Tours: Go Pro captured the rays under water" is retrieved, where the "ray" in the caption

---

[4]See Table 3 in Appendix A.

Table 8: Instances of the captions retrieved by our REPE on CIFAR100.

| Class | Retrieved captions |
|---|---|
| apple | "Apple slices stacked on top of each other"
"Apples growing on a tree"
"Still life with apples in a basket" |
| woman | "Portrait of a young woman"
"Woman standing at the window"
"Confident woman in a red dress and gold crown" |
| bridge | "The golden bridge in Bangkok"
"Bridge on the River Kwai ∼Video Clip"
"Wooden bridge over a mountain river" |
| ray | "Stingray in the Grand Cayman, Cayman Islands stock photography"
"Common Stingray swimming close to the sea floor."
"Sun Rays Tours: Go Pro captured the rays under water" |

Table 9: Accuracy of CLIP-Adapter and our REPE method on CIFAR100 and ImageNet datasets with few-shot learning.

| Method | K-shot | CIFAR100 | ImageNet |
|---|---|---|---|
| CLIP-Adapter | 4 | 66.6 | 63.0 |
| CLIP-Adapter + REPE | 4 | **67.5** $_{(+0.9)}$ | **63.3** $_{(+0.3)}$ |
| CLIP-Adapter | 16 | 69.0 | 64.6 |
| CLIP-Adapter + REPE | 16 | **69.8** $_{(+0.8)}$ | **64.9** $_{(+0.3)}$ |

refers to a narrow beam of light, heat, etc. We leave the retrieval with better semantics preservation for future work.

## H    COMPUTATIONAL OVERHEAD OF REPE

The computational overhead of REPE is three-fold (measured a single RTX TITAN GPU): **(1)** Computing the text and image embeddings of CC12M dataset (Changpinyo et al., 2021) we used. It takes about **an hour** with a single RTX TITAN GPU. **(2)** Building indices out of the embeddings for KNN search with the FAISS framework (Johnson et al., 2019). The cost depends on the hyper-parameter setting for the target recall, and the procedure can be finished in **an hour** for our dataset. **(3)** Retrieving the relevant images and corresponding captions for given query images. Once the indices are built, we can efficiently search over the dataset according to the query image in real time (less than **5 ms** for retrieving the top-100 relevant image-caption pairs), which is thus still applicable for query-intensive scenarios.

## I    INCORPORATION OF REPE TO DOWNSTREAM FINE-TUNING

Our REPE method is model-agnostic and orthogonal to parameter-tuning methods. We can combine it with fine-tuning methods like adapter-tuning (Gao et al., 2021) (denoted as CLIP-adapter), to achieve further performance boost. Concretely, CLIP-adapter adds a tunable adapter network (two linear layers with ReLU) after the last layer of both image and text encoder. Based on it, we ensemble the original class description ("a photo of a [CLASS]") with the captions retrieved by our REPE as the final textual input. The CLIP-Adapter is trained with K-shot training samples and then evaluated on full test splits. We conduct experiments on CIFAR100 and ImageNet with K=4 and K=16, respectively. As shown in Table 9, REPE further improves the performance of adapter-tuning by an average of 0.6, which demonstrates the adaptability and superiority of our method.

Table 10: A comparison of Closed Set Recognition, Open Set Recognition (OSR), Open World Recognition, and Open-vocabulary Recognition (OVR).

| Task | Paradigm | Goal | Signal | Training | Testing |
|------|----------|------|--------|----------|---------|
| Closed Set Recognition | Classification | Identifying known classes | Supervised | Known classes | Known classes |
| Open Set Recognition | Classification | Identifying known classes & rejecting unknown classes | Supervised | Known classes | Known classes & unknown classes |
| Open World Recognition | Classification | Identifying known classes & detecting unknown classes & labeling unknown data & incrementally learn and extend classifier | Supervised | Incremental known classes | Known classes & unknown classes |
| Open-vocabulary Recognition | Matching | Identifying classes via natural language | Unsupervised | - | Classes in a vocabulary |

## J  COMPARISON OF RELATED WORK

Table 10 provides a more detailed comparison of Closed Set Recognition, Open Set Recognition (OSR) (Scheirer et al., 2013; Geng et al., 2021), Open World Recognition (OWR) (Bendale & Boult, 2015), and Open-vocabulary Recognition (OVR) (Radford et al., 2021) from 5 perspectives paradigm, goal, signal, classes type in training, and classes type in testing, respectively. Contrary to other research, the CLIP-based Open-vocabulary Recognition is unsupervised. We focus on the model performance on zero-shot inference, without training or fine-tuning on the target dataset.

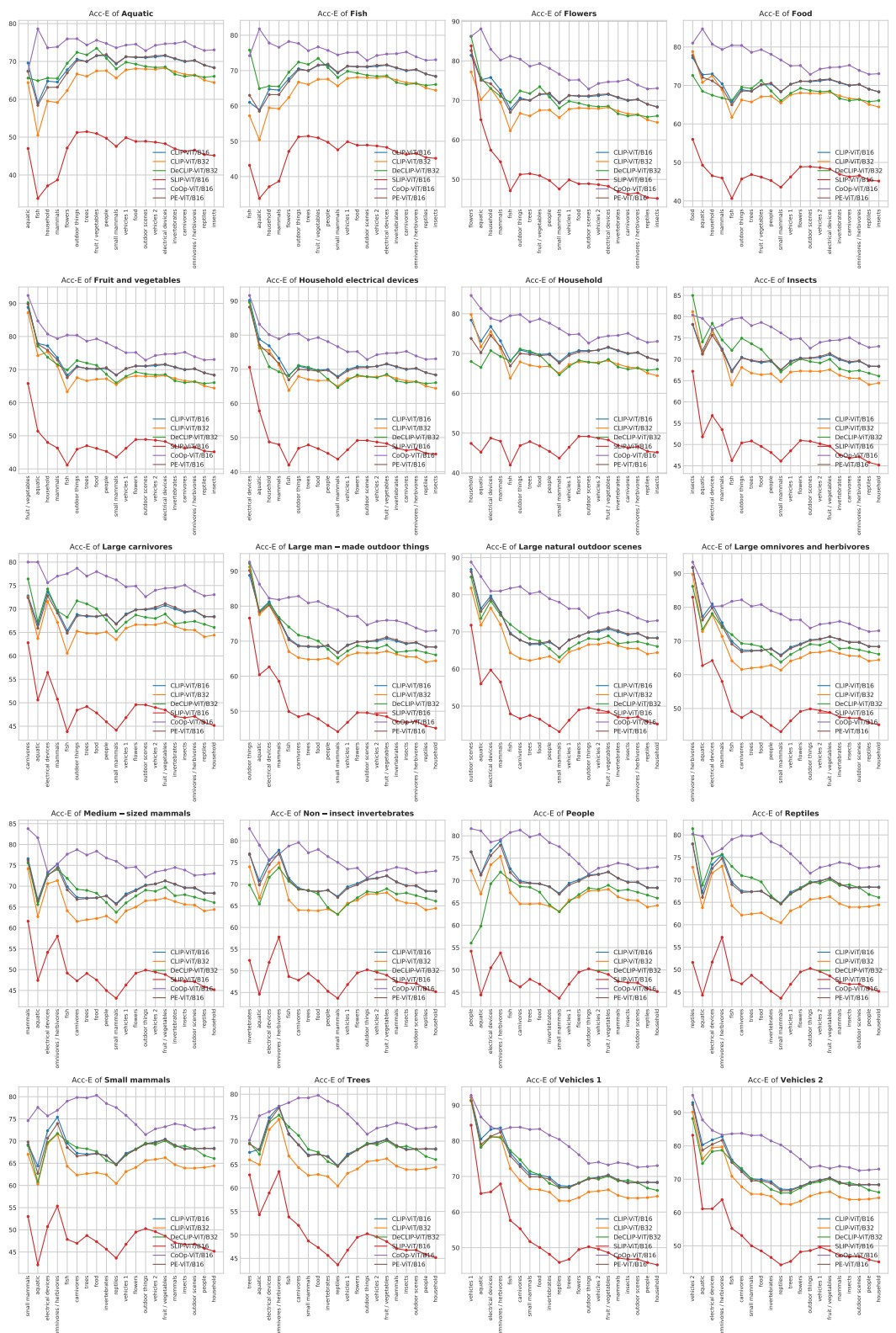

Figure 8: Incremental Acc-E of CLIP and its variants on CIFAR100.

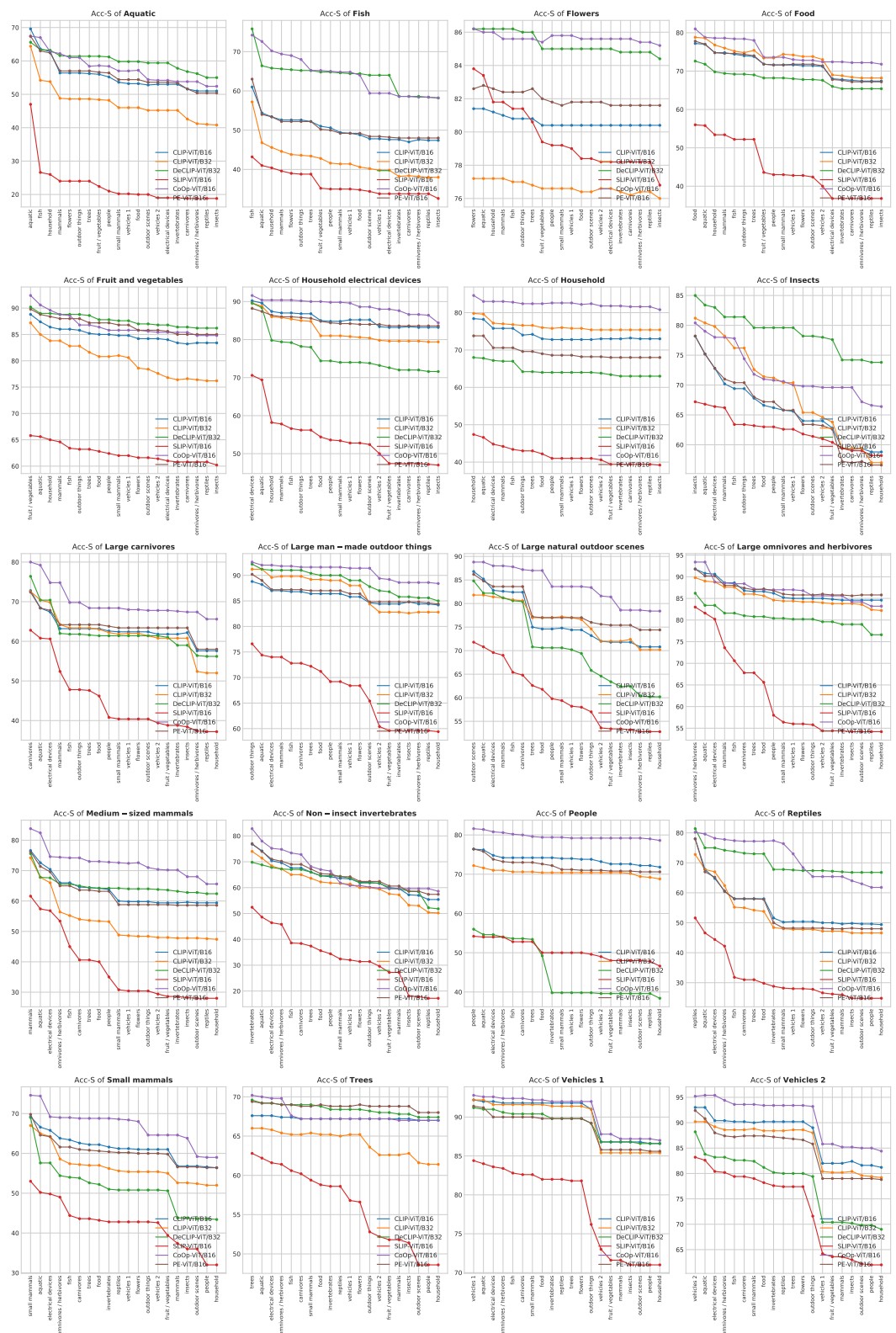

Figure 9: Incremental Acc-S of CLIP and its variants on CIFAR100.

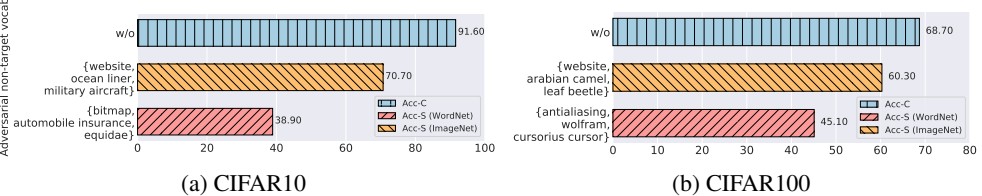

(a) CIFAR10                    (b) CIFAR100

Figure 10: Adversarial non-target vocabulary and the corresponding Acc-S on CIFAR datasets. Adding 3 non-target classes into the candidates leads to severe performance deterioration, revealing the vulnerability of CLIP when faced with malicious vocabulary.

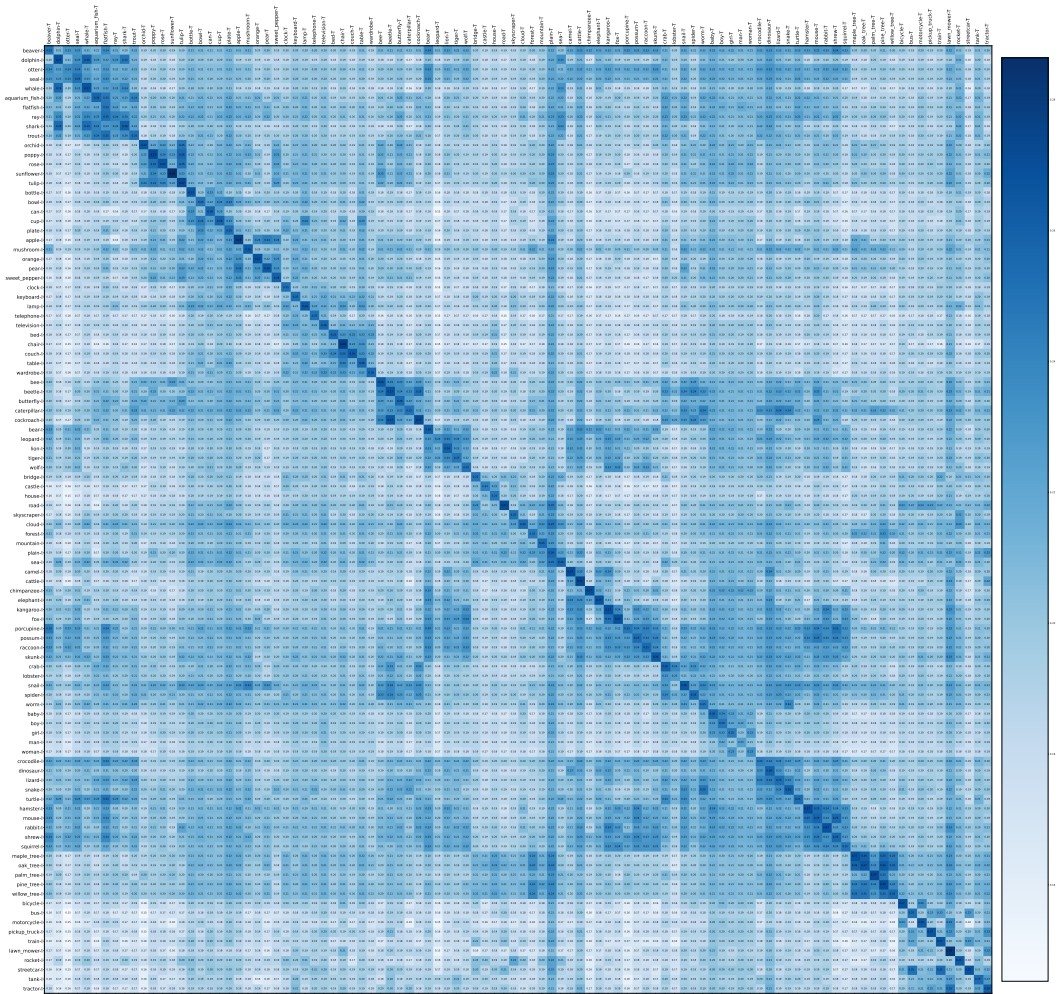

Figure 11: Averaged cosine similarity between image (-I) and prompt (-T) features of CLIP (ViT-B/32) on intact CIFAR100. The elements on the diagonal represent the similarity of the positive image-text pairs, while the others represent that of the negative ones.

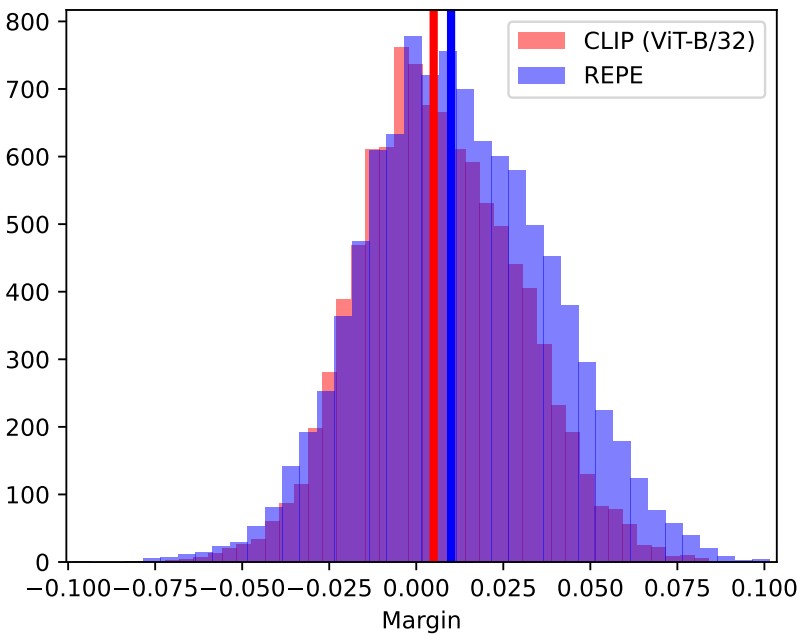

Figure 12: Margin distribution of similarity scores of our REPE (blue) and CLIP (ViT-B/32) (red). The median value of REPE's distribution is .01 (the blue vertical line), which is larger than that of CLIP (ViT-B/32) with .005 (the red line). It indicates that the predictions of REPE are harder to be inverted with competing classes than the original CLIP.

