# OpenReview forum: "Delving into the Openness of CLIP"
_ICLR.cc/2023/Conference — Submitted to ICLR 2023_

### Official Review · Reviewer_hc5X · 2022-10-25

**Confidence:** 4
**Clarity, Quality, Novelty And Reproducibility:** The explanation of the REPE approach …
**Correctness:** 1
**Technical Novelty And Significance:** 2
**Empirical Novelty And Significance:** 2
**Recommendation:** 5

**Strength And Weaknesses:**

**Strengths**

- The authors provide in-depth empirical study to analyze the performance of CLIP on novel concepts.

- The authors are studying an interesting area that is relevant and useful to the community.


**Weaknesses**

- The main weakness of that the paper is that it is very confusing and does not flow correctly. Unfortunately, I think it is because there is too much packed in. The first half of the paper addresses the problem the authors are trying to solve in great detail but was confusing because some terms were undefined and the purpose of this analysis was not necessarily clear. How does it differ from text-to-image retrieval? Is this all zero-shot, in which case, why are we interested if the model performs less accurately on a fixed, target dataset if we add some classes not relevant to the target dataset? How was the superclass hierarchy used? This is all a lot of experimentation but not clearly laid out.

- Then the last ¼ of the paper is a proposed approach and it felt very quick and rushed. It was very difficult to get the full grasp of the change or the approach that was used.

**More details:**

  - Terms are not properly defined, like “vocabulary extension”. This could mean many things, for example, the actual text-encoder has an embedding space with a fixed vocabulary size, is this what the authors are extending? If so, should models using a BERT-like text encoder not be compared since it uses sub-words during tokenization?

  - It is confusing how this task varies from text-to-image/image-to-text retrieval when you have an image and several text pairs and you are measuring the accuracy of that retrieval. Is this task not just taking a classification problem and making it more like a retrieval problem because the classes are no longer fixed? If the initial analysis is zero-shot, than this difference should probably be discussed as it would also evaluate the “openness” without claiming that the model performs poorly on this fixed classification datasets. Furthermore, given this is zero-shot learning, how is that not evaluating “oppenness” if it has not seen these images and text pairs during training? These concepts should be clarified and explained.

  - Using some kind of hierarchical superclass organization sounds very interesting but is not clearly explained.

  - The REPE approach definition mentions KNN and FAISS in one sentence, but does not provide enough information so it is confusing how this actually works.



**Summary Of The Paper:**

The goal of this study is to measure the “oppeness” of CLIP-based models. The authors are adding class names that are not part of the target dataset domain and evaluating the change in accuracy from that. They then show how poorly models do and propose a new approach to this via two metrics: inter-modal alignment and intra-modal uniformity.The authors also perform prompt-engineering by generating a better search algorithm from the original dataset.

**Summary Of The Review:**

Overall, it felt like there is too much focus on the underlying problem and not enough on their proposed approaches REPE. The reader is left with only questions.

If the paper is re-focused with more emphasis on the why and how this differs from other prompting approaches, and certain concepts clarified, this paper would have more clarity.

---

> ### Author Response · Authors · 2022-11-12
> **Responses to Reviewer hc5X (Part 1)**
>
> Thank you for the helpful feedback and the suggestions on writing. Our paper focuses on the area of **open-vocabulary visual recognition**, which has become popular in recent years after CLIP was proposed. Although we have introduced the overall framework and related work of CLIP in Sec. 1 and Sec. 4, we believe that there are some misunderstandings due to the omission of some preliminary information. **We will re-explain the central concept of CLIP and highlight the advancements of our evaluation protocol and REPE method.**
>
> > **Q1:** What is the difference between the task in the paper and the task of image-to-text retrieval?
>
> **A1:** The general task in our paper is **image classification**. Originally, it takes only images as input, and the target is a semantically meaningless one-hot label, and thus is completely different from the task of image-to-text retrieval.
>
> However, **CLIP reformulates the image classification task in an image-text retrieval style**. It learns to map an image to a textual class description (e.g., “a photo of a [CLASS NAME]”) instead of a discrete label ID. Given a downstream task with N classes, CLIP generates N class features by feeding each class description to the text encoder, then uses the input image to retrieve the most similar class feature and takes the corresponding class as the prediction.
>
> The advantage of such a modeling paradigm is that it can learn the visual-semantic matching based on web-scale collections of image-text pairs in a self-supervised way (i.e., contrastive vision-language learning), while the traditional paradigm requires a lot of human effort to label samples for each class (e.g., ImageNet). Most importantly, due to the flexibility of natural language, **we can directly generate text features for arbitrary classes in arbitrary downstream datasets, which enables the model to conduct zero-shot classification**. In contrast, in the traditional paradigm, the models need to re-train their classifiers to adapt to different datasets or newly emerged target classes, since the label ID (e.g., “0”) in different datasets represents different classes. Therefore, compared to CLIP, the traditional models are NOT intrinsically “open” to the real world.
>
> > **Q2:** What are “vocabulary” and “vocabulary extension”?
>
> **A2:** The term “vocabulary” in our area of **open-vocabulary image classification** is different from that in the area of NLP. In classic NLP, it refers to a set of words/sub-words/tokens (e.g., the vocabulary of BERT-Base-Uncased has 30522 tokens), and is mainly used in the tokenization phase. However, in open-vocabulary image classification, **the term “vocabulary” refers to the set of class names in downstream classification datasets** (e.g., the vocabulary of CIFAR10 has 10 items, including its class names `airplane`, `automobile`, etc.). In this scenario, it doesn’t matter if the items (i.e., class names) in the vocabulary have multiple words, because we only use the class names to write the textual class descriptions (e.g., “a photo of a [CLASS NAME]”), then feed them into the text encoder to generate class features for the classifications.
>
> Accordingly, **the term “vocabulary extension” refers to enlarging the class set by adding new classes to it**. In practice, we achieve “vocabulary extension” by combining two vocabularies. For example, in the upper right of Fig 1, the vocabulary $V_1^{(T)}$={“cat”, “dog”} and $V_2^{(T)}$={“tiger”, “lion”}, we can extend $V_1^{(T)}$ by merging $V_2^{(T)}$ into it and then obtain $V_1^{(T)} \cup V_2^{(T)}$={“cat”, “dog”, “tiger”, “lion” }.
>
> Note that in our paper, **the terms “(target) vocabulary” and “vocabulary extension” are defined in Sec. 2.1 and Definition 2.1, respectively**.
>
> > **Q3:** How is the superclass-class hierarchical structure used?
>
> **A3:** As we introduced in Sec. 2.2, **our evaluation protocol requires N vocabularies** to calculate the extensibility and stability metrics. Instead of constructing a new benchmark from scratch, **we reuse the existing downstream datasets** (CIFAR100, ImageNet), and **uniformly split their class set into N groups (i.e., N vocabularies). The principle of class set splitting is the underlying superclass-class hierarchical structure of the datasets**, which was predefined in previous work. [1,2]
>
> Based on such a structure, the classes belonging to the same superclass will be grouped into a vocabulary (Please refer to the examples in Appendix A). It allows us to quantify the model's capability in dealing with recognition targets from a newly emerged superclass and can aid in analyzing the compatibility of different classes and superclasses.
>
> [1] Learning multiple layers of features from tiny images. Krizhevsky & Hinton, 2009
>
> [2] BREEDS: benchmarks for subpopulation shift. Santurkar et al., 2021

---

> > ### Author Response · Authors · 2022-11-12
> > **Responses to Reviewer hc5X (Part 2)**
> >
> > > **Q4:** The vanilla evaluation mechanism can also reflect the openness of CLIP since the evaluation is done in a zero-shot manner.
> >
> > **A4:** Yes, CLIP's original paper reported its impressive zero-shot performance on multiple downstream datasets. This indeed reflects the openness of CLIP. As we quote in our paper, **CLIP is open to arbitrary vocabularies operationally, which is the basis of our paper**.
> >
> > However, what we argue is that **the vanilla evaluation mechanism is static and restricted, leading to an overestimation of CLIP's openness**. For example, according to the vanilla evaluation setup, the average score of CLIP (ViT-B/32) on CIFAR100, Caltech101, and SUN397 is 72% (65%, 89%, and 62%, respectively). However, as shown in our evaluation for dataset-level vocabulary expansion (Appendix B), when we union the input and label sets of these three datasets, the overall accuracy dropped dramatically to 62%. For CIFAR100, an additional 14% of wrong predictions arose after the label set union (reducing the accuracy from 65% to 51%).  As a result, simply averaging the accuracy across multiple datasets cannot accurately reflect the actual openness.
> >
> > Accordingly, we take a further step to propose a new evaluation protocol. **Our protocol has three advantages** over static and restricted evaluation:
> > 1. **Our protocol explicitly models the dynamics of the real open world, and formulates the empirical risk of CLIP when new vocabularies incrementally emerge**. The assessment is standardized and multi-granularity. In Sec 2, the vocabularies are constructed via an internal superclass-class hierarchy within a dataset, which keeps the data distribution of each vocabulary. In Appendix B, the vocabulary expansion is conducted at the dataset level, which corresponds to a wild scenario.
> > 2. **Our protocol is designed in an incremental style, which helps us investigate the model’s behavior under increasing demand for openness**. From the curve of the accuracy during N times vocabulary expansion (Fig 2a, Fig 8, and Fig 9), we can grasp the trend of the models’ actual openness. From the adversarial vocabulary mining (Appendix D), we can explore the lower bound of the stability of CLIP in the open world.
> > 3. **Our protocol provides a tool to analyze the compatibility between different classes/vocabularies**. For example, as shown in Sec 2.3, CLIP performs stably on the vocabulary `Flowers` while is brittle for the vocabulary `Medium-sized Mammals`. It indicates that the `Flowers` is compatible with other vocabularies, incorporating classes from `Flowers` will not influence the model prediction on other classes, or vice versa. However, for `Medium-sized Mammals`, its co-occurrence with other vocabularies can lead to a large drop in model performance.

---

> > > ### Author Response · Authors · 2022-11-12
> > > **Responses to Reviewer hc5X (Part 3)**
> > >
> > > > **Q5:** What is the difference between REPE and previous prompting methods? How do KNN and FAISS work in the REPE approach?
> > >
> > > **A5:** Previous prompting methods either write the prompt manually (e.g., “a photo of a [CLASS]”) or fine-tune a soft prompt using downstream task datasets (CoOp [3]). **Their prompts share the same context for every class name** (e.g., for the class names “apple” and “bridge” in CIFAR100, the class descriptions are “a photo of an apple” and “a photo of a bridge,"  respectively), which cannot holistically represent the semantics of different visual categories. Instead, in REPE, **we customize the text prompt for each class with diverse captions from the pre-training corpus**. Specifically, for the class names “apple” and "bridge", the retrieved captions include “Apples growing on a tree” and “Wooden bridge over a mountain river." respectively (see the case study in Appendix G). Conducting the prompt ensemble with these captions **makes the text features of different classes more distinguishable, and thus can improve the model's performance.**
> > >
> > > To achieve this, we use CLIP to retrieve corresponding captions for each class. Since CLIP is a cross-modal retrieval model, we first take the text prompt “a photo of a [CLASS]” as a query to recall the most similar images. Then we ensemble the paired captions of the retrieved images to enhance the class description.
> > >
> > > Technically, we first compute the text and image embeddings of all the caption-image pairs in the pre-training dataset (e.g., the CC12M dataset) to form a database. To find the top-k similar images for a given text query, **a naive solution is an exact KNN search**. It conducts an exhaustive search by computing the vector similarity of the query vector (i.e., the text embedding of the description), with every image embedding in the database. This approach is computation-intensive for databases of our scale, and the delay could be intolerable. Therefore, we implement REPE with **FAISS, an efficient vector retrieval framework, with various state-of-the-art KNN indexing algorithms**. FAISS takes care of the computation complexity and search recall, by partitioning the index into cells and utilizing product quantization to speed up retrieval (we refer the reviewer to the FAISS paper [4] for more technical details).  **With FAISS, we can efficiently search over the dataset according to the query image in real time**, e.g., in less than 5 ms for retrieving the top-100 relevant image-caption pairs.
> > >
> > > We will incorporate this in the revision to help readers understand REPE better.
> > >
> > > [3] Billion-scale similarity search with GPUs, https://arxiv.org/pdf/1702.08734.pdf
> > >
> > > [4] Learning to Prompt for Vision-Language Models, https://arxiv.org/abs/2109.01134
> > >
> > > **We hope our response can address your concerns and we would appreciate it if you could reassess the paper.**

---

### Official Review · Reviewer_c4Um · 2022-10-25

**Confidence:** 4
**Correctness:** 2
**Technical Novelty And Significance:** 3
**Empirical Novelty And Significance:** 3
**Recommendation:** 5

**Clarity, Quality, Novelty And Reproducibility:**

The paper is well written and the code has been released so there is no issue to reproduce the paper. There are lots of experiments in the paper, yet the experiment design is not reasonable to validate the assumption.

**Strength And Weaknesses:**

The paper is easy to understand and the topic of evaluating the openness of CLIP in visual task is good and important. However there are some concerns for me to accept the paper.
- For Sec 2.2, the author proposed ACC-E to validate the extensibility of the CLIP by evaluate on different permutation of the vocabulary set.  I do not fully understand the principle and advances of the proposed evaluation mechanism. A more straightforward way is to apply the CLIP directly in different datasets with different vocabulary sizes and report the average score, especially for some uncommon classes which may not been covered by the pertaining stage of CLIP, e.g., the local food dishes in different countries.
- For Sec 2.2 and Table 1, the openness issue is reported in the small dataset CIFAR100. With larger dataset imageNet and stronger CLIP backbone, the gap between Acc-E/Acc-S and Acc-C reduces significantly (~13% -> ~3%). In real world, the class number is far more than ImageNet and the gap may potentially become even more marginal.
- For Sec 2.3, it's quite reasonable in visual recognition task that involving more classes, the overall performance will drop due to the class confusion.
- For Table 2, the results fail to validate the effectiveness of REPE to address the openness issue. The gap between Acc-E/Acc-S and Acc-C remains unchanged for both vanilla prompt design and REPE, except REPE reports higher results. REPE is a prompt engineering module and thus it's reasonable the results can improve. However, the main topic of the paper is not the prompt engineering but the openness issue of CLIP, and this issue still remains.

**Summary Of The Paper:**

The paper introduced a new concept extensibility, to explore the openness of CLIP on visual recognition task. Two new evaluation metrics Acc-E and Acc-S were proposed to quantify CLIP's extensibility and stability. The paper argued the confusion between text embeddings of different classes made the CLIP unstable in extensibility, and finally a new prompt engineering module, REPE was proposed to handle this issue.

**Summary Of The Review:**

The motivation of the paper is important, to evaluate the openness of CLIP in visual task. However, the experiment design is not reasonable and the proposed prompt engineering cannot address reported issue in the paper either. Overall speaking, the paper is below the bar of acceptance.

---

> ### Author Response · Authors · 2022-11-12
> **Responses to Reviewer c4Um (Part 1)**
>
> Thank you for giving us constructive feedback. As you said, the openness issue we investigated in our paper is crucial. We will clarify the advance of our proposed evaluation mechanism, and rejustify the effectiveness of our REPE method in addressing the openness issue.
> > **Q1: What are the principles and advances of the proposed evaluation mechanism (extensibility and stability)? A more straightforward way to evaluate the openness of CLIP is by directly applying it to different datasets with different vocabulary and reporting the average score.**
>
> **A1:** The more straightforward evaluation mechanism you mentioned is adopted in the original CLIP paper. The paper reported the impressive zero-shot performance of CLIP on multiple downstream datasets, including uncommon and fine-grained vocabularies/classes (such as FGVC Aircraft and Stanford Cars). This indeed reflects the openness of CLIP. As we quote in our paper, **CLIP is open to arbitrary vocabularies operationally, which is the basis of our paper**.
>
> However, what we argue is that **the vanilla evaluation mechanism is static and restricted, leading to an overestimation of CLIP's openness**. For example, according to the vanilla evaluation setup, the average score of CLIP (ViT-B/32) on CIFAR100, Caltech101, and SUN397 is 72% (65%, 89%, and 62%, respectively). However, as shown in our evaluation for dataset-level vocabulary expansion (Appendix B), when we union the input and label sets of these three datasets, the overall accuracy dropped dramatically to 62%. For CIFAR100, an additional 14% of wrong predictions arose after the label set union (reducing the accuracy from 65% to 51%).  As a result, simply averaging the accuracy across multiple datasets cannot accurately reflect the actual openness.
>
> Recall that in the real world, visual concepts are myriad, and new categories emerge constantly. A truly-opened visual recognition system is required to identify the target from all potential classes, which are even dynamically updating and scaling. How does the zero-shot CLIP perform when the label set is further expanded until it approaches the “full set”? How stable is the model's prediction for the old vocabularies (e.g., CIFAR100) during such expansion? What kinds of vocabulary updates and expansions are compatible with the model, and what kinds are dangerous? **These are the questions we care about but cannot be answered by previous evaluation mechanisms.**
>
> Accordingly, we take a further step to propose a new evaluation protocol. **Our protocol has three advantages** over static and restricted evaluation:
> 1. **Our protocol explicitly models the dynamics of the real open world, and formulates the empirical risk of CLIP when new vocabularies incrementally emerge**. The assessment is standardized and multi-granularity. In Sec 2, the vocabularies are constructed via an internal superclass-class hierarchy within a dataset, which keeps the data distribution of each vocabulary. In Appendix B, the vocabulary expansion is conducted at the dataset level, which corresponds to a wild scenario.
> 2. **Our protocol is designed in an incremental style, which helps us investigate the model’s behavior under increasing demand for openness**. From the curve of the accuracy during N times vocabulary expansion (Fig 2a, Fig 8, and Fig 9), we can grasp the trend of the models’ actual openness. From the adversarial vocabulary mining (Appendix D), we can explore the lower bound of the stability of CLIP in the open world.
> 3. **Our protocol provides a tool to analyze the compatibility between different classes/vocabularies**. For example, as shown in Sec 2.3, CLIP performs stably on the vocabulary `Flowers` while is brittle for the vocabulary `Medium-sized Mammals`. It indicates that the `Flowers` is compatible with other vocabularies, incorporating classes from `Flowers` will not influence the model prediction on other classes, or vice versa. However, for `Medium-sized Mammals`, its co-occurrence with other vocabularies can lead to a large drop in model performance.

---

> > ### Author Response · Authors · 2022-11-12
> > **Responses to Reviewer c4Um (Part 2)**
> >
> > > **Q2: The openness issue on datasets with a larger class number (e.g., ImageNet) reduces significantly compared to that on CIFAR100 (from 9.8% to 2.8% on average).**
> >
> > **A2:** Not really. The openness issue is not simply correlated with the class number or the size of the dataset. The following table shows CLIP performance on another vocabulary split (Mixed13 [1]) for ImageNet. **The openness issue** (the gap ($\Delta$) between Acc-C and Acc-E/Acc-S) **on this split is almost as serious as that on CIFAR100** (average gap: **9.4% vs. 9.8%**).
> >
> > As we addressed in Secs. 2.3 and 5, the openness issue is influenced by the inherent properties of the image categories, such as their degree of abstraction, i.e., whether we can easily describe them in natural language, and their concept frequency in the training dataset. **In short, the openness issue is serious and much more complicated, and are worth further study.**
> > | Model          |   [    |       | ImageNet (Mixed13) |       |         |   ][   |          |          | CIFAR100  |          |       ]    |
> > |-----------------|:-----:|:-----:|:------------------:|:-----:|:--------:|:---:|:--------:|:--------:|:---------:|:--------:|:---------:|
> > |                 | Acc-C | Acc-E |      $\Delta$      | Acc-S | $\Delta$ |     |  Acc-C   |  Acc-E   | $\Delta$  |  Acc-S   | $\Delta$  |
> > | CLIP (RN101)    | 98.1  | 88.9  |        -9.2        | 88.2  |   -9.9   |     |   68.3   |   55.4   |   -12.9   |   54.9   |   -13.4   |
> > | CLIP (ViT-B/32) | 98.4  | 89.0  |        -9.4        | 88.3  |  -10.1   |     |   78.0   |   69.6   |   -8.4    |   68.9   |   -9.1    |
> > | CLIP (ViT-B/16) | 98.4  | 90.1  |        -8.3        | 89.2  |   -9.2   |     |   79.7   |   72.6   |   -7.1    |   72.0   |   -7.7    |
> >
> > Vocabulary split for ImageNet (Mixed13):
> > | Vocabulary |                                                 Classes                                                  |
> > |:----------:|--------------------------------------------------------------------------------------------------------|
> > |   mix 1    |        horse chestnut seed, leaf beetle, garbage truck, barber chair, pickup truck, snow leopard         |
> > |   mix 2    |                          acorn squash, lynx, sea lion, ambulance, ostrich, jeep                          |
> > |   mix 3    |              black-and-white colobus, fire truck, ground beetle, Shih Tzu, sailboat, agaric              |
> > |   mix 4    |                goldfinch, hammerhead shark, station wagon, Pekingese, house finch, tench                 |
> > |   mix 5    |         small white butterfly, chiffonier, china cabinet, convertible, rooster, longhorn beetle          |
> > |   mix 6    |              electric ray, hand-held computer, stinkhorn mushroom, guenon, jaguar, macaque               |
> > |   mix 7    |                    hen, leopard, dung beetle, police van, goldfish, earth star fungus                    |
> > |   mix 8    |               slide rule, Maltese, Saharan horned viper, tow truck, tiger beetle, website                |
> > |   mix 9    |                rapeseed, lifeboat, notebook computer, cougar, Granny Smith apple, gondola                |
> > |   mix 10   |                   ladybug, bookcase, canoe, gyromitra, laptop computer, Japanese Chin                    |
> > |   mix 11   |             fireboat, desktop computer, chiton, brambling, hen of the woods mushroom, langur             |
> > |   mix 12   | moving van, Chihuahua, desert grassland whiptail lizard, great white shark, King Charles Spaniel, baboon |
> > |   mix 13   |                 tiger shark, coral fungus, patas monkey, motorboat, bassinet, limousine                  |
> >
> > [1] https://github.com/MadryLab/robustness
> >
> > > **Q3: It’s not surprising that performance drops as additional image classes are added.**
> >
> > **A3:** Intuitively, we can indeed expect the performance to drop as the vocabulary expands, **yet a systematic evaluation is still needed to analyze the reason behind the performance drop.**
> > To the best of our knowledge, we are among the first to set up such a formulation to quantify the capability of the models for real-life open tasks and provide a tool for probing the degraded performance during vocabulary expansion.
> >
> > Besides, based on our definition of stability, **it is surprising to find that only three maliciously-selected non-target classes (adversarial vocabulary in Appendix D) could lead to an absolute 52.7% accuracy drop on CIFAR10**, indicating that CLIP exhibits a poor modeling capability for objects with high abstraction levels. This phenomenon cannot be simply attributed to the expansion of vocabulary, and we hope our findings can facilitate future studies investigating this phenomenon.

---

> > > ### Author Response · Authors · 2022-11-12
> > > **Responses to Reviewer c4Um (Part 3)**
> > >
> > > > **Q4: The results in Table 2 fail to validate the effectiveness of REPE to address the openness issue since the gap between Acc-E/Acc-S and Acc-C remains unchanged.**
> > >
> > > **A4:** Not really. We provide the gap between Acc-E/Acc-S and Acc-C in the following table. For vanilla CLIP, the average gap is **15.3**, while **for REPE, the average gap is narrowed to 14.0**. It indicates that REPE not only improves the three metrics (i.e., Acc-C, Acc-E, and Acc-S), **but also reduces the performance drop when new classes are added.**
> > >
> > > Furthermore, Fig 12 in Appendix E demonstrates the margin distribution of vanilla CLIP and our REPE (ViT-B/32). Compared to vanilla CLIP, **REPE pushes the overall distribution towards the positive side and doubles the median value of the margin distribution from 0.005 to 0.01.** It indicates that REPE enlarges the margin between positive and negative class features, and thus the predictions are harder to be inverted with competing classes.
> > >
> > > **In summary, all these results validate the effectiveness of REPE in addressing the openness issue.**
> > >
> > > | Model           |   [       |          | CIFAR100  |          |           |   ][       |          | ImageNet (Entity13) |             |            |     ][     |              | ImageNet (Living17) |              |       ]     |
> > > |-----------------|:--------:|:--------:|:---------:|:--------:|:---------:|:--------:|:--------:|:-------------------:|:-----------:|:----------:|:--------:|:------------:|:-------------------:|:------------:|:----------:|
> > > |                 |   Acc-C  |  Acc-E   | $\Delta$  |  Acc-S   | $\Delta$  |  Acc-C   |  Acc-E   |      $\Delta$       |    Acc-S    |  $\Delta$  |  Acc-C   |    Acc-E     |      $\Delta$       |    Acc-S     |  $\Delta$  |
> > > | CLIP (RN101)    |   68.3   |   55.4   |   -12.9   |   54.9   |   -13.4   |   80.4   |   77.4   |        -3.0         |    77.3     |    -3.1    |   77.6   |     74.5     |        -3.1         |     74.4     |    -3.2    |
> > > | REPE (RN101)    | **68.4** | **55.5** | **-12.9** | **55.2** | **-13.2** | **81.7** | **79.2** |      **-2.5**       |  **79.0**   |  **-2.7**  | **77.8** |   **75.3**   |      **-2.5**       |   **75.2**   |  **-2.6**  |
> > > | CLIP (ViT-B/32) |   78.0   |   69.6   |   -8.4    |   68.9   |   -9.1    |   80.8   |   78.0   |        -2.8         |    77.8     |    -3.0    |   78.0   |     74.4     |        -3.6         |     75.0     |    -3.0    |
> > > | REPE (ViT-B/32) | **78.5** | **70.9** | **-7.6**  | **70.6** | **-7.9**  | **82.3** | **79.8** |      **-2.5**       |  **79.6**   |  **-2.7**  | **79.0** |   **76.4**   |      **-2.6**       |   **76.2**   |  **-2.8**  |
> > > | CLIP (ViT-B/16) |   79.7   |   72.6   |   -7.1    |   72.0   |   -7.7    |   83.5   |   81.1   |        -2.4         |    81.0     |    -2.5    |   79.5   |     77.9     |        -1.6         |     77.6     |    -1.9    |
> > > | REPE (ViT-B/16) | **79.8** | **72.9** | **-6.9**  | **72.6** | **-7.2**  | **85.4** | **83.3** |      **-2.1**       |  **83.2**   |  **-2.2**  | **79.9** |   **78.4**   |      **-1.5**       |   **78.2**   |  **-1.7**  |
> > >
> > > (For Acc-C, Acc-E, Acc-S, and $\Delta$, higher is better.)
> > >
> > > **We hope we have addressed your concerns. Discussions are always open. Thank you!**

---

### Official Review · Reviewer_8DmZ · 2022-10-26

**Confidence:** 5
**Correctness:** 4
**Technical Novelty And Significance:** 2
**Empirical Novelty And Significance:** 2
**Recommendation:** 5

**Clarity, Quality, Novelty And Reproducibility:**

The writing quality of the paper is clear and the paper is quite easy to follow.
The authors include detailed explanations that are intuitive and the motivation of the paper is very clearly defined and the paper is well-argued. The presentation is of high-quality.
The technical novelty, as mentioned in the weaknesses above, is quite low. The significance of the method or contribution is not clear. In my view, it is of incremental novelty over existing prompt leaerning approaches.
The paper is also reproducible - sufficient details are present to clearly implement the approach.

**Strength And Weaknesses:**

[Strengths]
The authors do a deep-dive on an important aspect of these large-scale vision-language models, i.e. how well do they actually generalize to their use-case. To do this, they examined two measures to account for this performance drop in the real-world - i.e. extensibility and stability. They also do a deep-drive into the feature spaces of these various methods using a number of metrics they develop and demonstrate quantitatively the problems with the feature spaces that their method tries to address.

Beyond just evaluating the method and exposing the problem on CLIP's feature space, the authors also propose a method to improve CLIP's extensibility and stability. The proposed method is quite straightforward but generates consistent small gains over not using it.

[Weaknesses]
As the authors cite in their paper, much past work has explored prompt tuning for CLIP. For example, in Table 1, we see CoOP (a prompt tuning method) achieves 76.7% while authors method, shown only with vanilla CLIP in Table 2 achieves 72.6%, for example. In other words, the CoOp approach works better than authors' method. Of course, CoOp requires access to the downstream target dataset. Yet, is this really a fair comparison, since in your approach we *do* have access to a dataset used to train CLIP. Why not use your idea to retrieve relevant images / text on the CLIP training dataset, if that's fair game?
In sum, the authors' approach is assuming that the large-scale CLIP pre-training dataset is available for querying. But if it is, perhaps some of these other methods could work better as well. In that scenario, it is unfair to compare just to vanilla CLIP.

The authors' method is also of low technical novelty and consists mainly of finding other captions in the train dataset and taking an average. Similarly, the proposed "measures" like the average distance between positive classes and the uniformity are quite simple. It is also unclear to me whether the intra-modal uniformity is a good measure. For example, it adds together the average distance between image pairs and text pairs. But these are two embedding spaces learned by two different models. Shouldn't these be normalized in some way so the two distributions (image and text embedding space) is directly comparable? For example, right now one modality might dominate. Thus, there is very little technical novelty to authors' approach.

Similarly, while authors have pointed out important problems, it isn't clear to me that the proposed approach is well-motivated to solving them. For example, why does taking the average caption solve these problems? Perhaps a better training paradigm, different representation space (e.g. hyperbolic), etc. is the correct way to address the problems the authors have pointed out, rather than just taking the average caption. Thus, it seems that the authors point out a clear problem, but that the proposed method seems ad-hoc and unrelated to the action problem. Perhaps a better way to handle it would be to learn some class-agnostic projection that solves these problems or something.

I'll also point out that the author's method requires this additional step of keeping the CLIP training dataset, indexing it, and doing this class-averaging step. Thus, there is this unnatural overhead to using their method for only around a 1.2% gain in performance. It's not clear that the overhead justifies its use.


**Summary Of The Paper:**

The authors perform a detailed study of the "openness" of the CLIP image-text model and variants. Specifically, they study the ability of CLIP to generalize to new categories of objects in the zero-shot setting. For this, they propose an evaluation protocol that analyzes CLIP's performance along two axes - extensibility and stability. Extensibility refers to the ability of the model to deal with new object classes when the vocabulary of object classes is expanded. This means, having a set of target classes which is progressively grown in size by adding new categories. The measure of stability is interesting as it refers to keeping the set of target classes fixed, but adding a number of distractor classes that the model could potentially classify, but for which no images are actually present. Models which are stable are able to continue performing well on the set of target classes despite the distractors. The authors argue that stability may more realistically account for the model's performance in the real-world zero-shot setting.
The authors results show that CLIP models suffer from poor extensibility and stability in the real world. They show significant performance drops using three benchmarks (CIFAR100, ImageNet (Entity13) and ImageNet (Living17).
To explain the performance drops, the authors study the margin between positive and negative classes and observe this is the direct contributing factor to the model's poor stability. The authors discover classes which, when added to the set of options, cause significant misclassifications based on the similarity of the features.
The authors also study two other properties of the feature space. Specifically, the inter-modal alignment and intra-modal uniformity.
Inter-modal alignment is essentially just the average distance between the positive image and text pairs of embeddings.
The intra-modal uniformity basically samples pairs of samples from the same modality and takes their distance.
They show that existing models perform poorly on these measures.
Finally, authors propose a new method for addressing these issues called REPE. The basic idea is to not use a standard text prompt like "a photo of a [class]" but to instead learn a class-specific text-embedding. To do this, given a prompt of the class name, the authors look in the original dataset used to train CLIP to find similar images. Then, they find the captions that correspond to those images. They encode the captions and then essentially just average them with some weighting.
The authors show that this approach improves the extensibility and stability of CLIP by 1.2% compared to not using it.

**Summary Of The Review:**

The authors' paper makes a compelling case by pointing out key weaknesses of CLIP. They demonstrate this both quantitatively and qualitatively through illustrations in their paper. They include many supplemental results as well. However, the proposed method seems ad-hoc and doesn't necessarily directly address the problems pointed out. It also is of low novelty and results in only minor performance gains with considerable technical overhead. Thus, in my view the paper should not be accepted at this time.

---

> ### Author Response · Authors · 2022-11-18
> **Responses to Reviewer 8DmZ (Part 1)**
>
> Thank you for giving us constructive feedback. We provide the following justification for the fairness of our experiments and the superiority of our REPE method.
>
> > **Q1: It is unfair to merely compare REPE to vanilla CLIP in Table2. The prompt tuning methods like CoOp should also be included, even if it requires access to the downstream dataset.**
>
> **A1:** Not really. We recall that, as stated in the Introduction, our evaluation protocol for openness derives from the vanilla CLIP evaluation protocol for zero-shot image classification. It means that **our evaluation also demands no access to any training data from downstream datasets or new emerging classes.** Therefore, it’s fair to compare REPE to vanilla since they are both blind to the target data distribution during the evaluation, and REPE only reuses some pre-training data points that have been seen by vanilla CLIP.
>
> As for CoOp, which leverages downstream datasets to fine-tune the parameters, we do not compare it to vanilla CLIP and REPE since it is really unfair. Despite this, **such a supervised model can be considered the upper bound of zero-shot models** (e.g., vanilla CLIP and REPE), so we report its performance in Table 1 and analyze its optimization trajectory in Fig 7 to find the key to addressing the openness issue (i.e., improving text feature distinguishability and semantic alignment).
>
> > **Q2: It is not clear whether the metric of intra-modal uniformity is good, because it simply adds up the average distance between image pairs and text pairs without normalizing their embeddings to make the two distributions comparable.**
>
> **A2:** Not really. All the image and text embeddings in our paper (including the calculation of the intra-modal uniformity) are L2-normalized. We omitted it since it is a default operation in CLIP. **Therefore, the two distributions of image and text embeddings are directly comparable, and the metric is technically sound.**
>
> Additionally, **our metrics**—inter-modal alignment and intra-modal uniformity—**are beneficial because they capture two essential characteristics of the optimal feature space in the learning of vision-and-language representations**. First (inter-modal alignment), the text feature of a class name is supposed to stay close to the features of the images it describes, promoting the similarity of positive pairs. Second (intra-modal uniformity), intra-modal features, especially the textual features should be uniformly distributed to preserve maximal information and make the descriptions of competing categories more distinguishable.
>
> > **Q3: For the openness pointed out by the authors, it is not clear that the proposed approach REPE is well-motivated to solve it, since REPE seems to only take the average caption.**
>
> **Q3:** Our REPE method is well-motivated to solve the openness issue. Our analysis of CLIP's margin distribution and representation space indicates that **the poor distinguishability of competing class features is the primary cause of the openness problem in CLIP-like models**.  Recall that the context for each class name is the same in vanilla CLIP-like models (e.g., “a photo of a [CLASS]”),  making it difficult to discriminate between distinct visual categories because the semantics of each cannot be fully represented.
>
> Instead, in REPE, we customize each class description with diverse captions from the pre-training corpus. Specifically,  the retrieved captions for the class names "apple" and "bridge" are, respectively, "Apples growing on a tree" and "Wooden bridge over a mountain river" (See the case study in Appendix G).  **These captions make it easier to identify between the text features of various classes when conducting the prompt ensemble, which can enhance model performance.**
>
> As shown in Table 2, REPE consistently improves the extensibility and stability of CLIP by an average of 1.2%. Additionally, REPE increases the median value of the distribution from 0.005 to 0.01 and pushes the margin distribution towards the positive side compared to vanilla CLIP (See Fig. 12 in Appendix E). It indicates that REPE widens the gap between positive and negative class features, making it more difficult to invert predictions with competing classes. In conclusion, **all of these findings support REPE's efficacy in addressing the openness problem.**

---

> > ### Author Response · Authors · 2022-11-18
> > **Responses to Reviewer 8DmZ (Part 2)**
> >
> > > **Q4: Compared to REPE, perhaps a better way to address the openness issue is to design a better pre-training paradigm.**
> >
> > **A4:** We appreciate your thoughtful advice. We agree that a better representation space derived from a better training paradigm can alleviate the openness issue. Actually, in Table 1, we have conducted an exhaustive evaluation on CLIP-like models with existing pre-training paradigms. We verified that introducing a stronger vision backbone (ViT v.s. ResNet), widespread supervision (DeCLIP v.s. CLIP), and more pre-training data (CLIP v.s. SLIP) can improve the extensibility and stability of models on open tasks, which is also one of our contributions.
> >
> > **We also concur that by utilizing the results of our analysis, we can further enhance the pre-training paradigm.** For example, we can explicitly enforce the model to optimize the losses of inter-modal alignment and intra-modal uniformity, or further enlarge the margin between the positive and negative similarity.
> >
> > However, with the help of our analytical findings, we have shown a simple and effective way (REPE) to improve the openness of CLIP. We do not conduct in-depth tuning for experimental results, as it is beyond the scope of this paper. The analysis findings in this work, in our opinion, are sufficiently significant.  **It will be an independent innovation and contribution to use the analysis findings of this research to suggest more successful method, thus we leave it to future work.**
> >
> > > **Q5: The REPE method requires the additional step of keeping the CLIP training dataset, indexing it, and doing this class-averaging step the performance is only around 1.2%. It is not clear that the overhead of REPE justifies its use.**
> >
> > **A5: First, the cost of REPE is affordable.** REPE does not require maintaining the raw pre-training dataset. We can just pre-encode all the images and captions and save them in the feature format. Even though the feature pre-encoding and indexing process takes roughly two hours, it only needs to be done once. Once complete, we can quickly use FAISS to search the dataset in accordance with the image query in real time, for example, taking less than 5 ms to retrieve the top 100 pertinent picture-caption pairs.
> >
> > **Second, we argue that improving the performance of zero-shot image classification is not trivial, and that our performance improvement is substantial and comparable to powerful pre-training techniques like DeCLIP:**
> > | Model             |    [      |          | CIFAR100 |          |         |    |      ] [     |          | ImageNet (Entity13) |          |          |    |      ][     |          | ImageNet (Living17) |          |      ]    |
> > |-------------------|:--------:|:--------:|:--------:|:--------:|:--------:|:---:|:--------:|:--------:|:-------------------:|:--------:|:--------:|:---:|:--------:|:--------:|:-------------------:|:--------:|:--------:|
> > |                   |  Acc-C   |  Acc-E   | $\Delta$ |  Acc-S   | $\Delta$ |     |  Acc-C   |  Acc-E   |      $\Delta$       |  Acc-S   | $\Delta$ |     |  Acc-C   |  Acc-E   |      $\Delta$       |  Acc-S   | $\Delta$ |
> > | CLIP (ViT-B/32)   |   78.0   |   69.6   |   -8.4   |   68.9   |   -9.1   |     |   80.8   |   78.0   |        -2.8         |   77.8   |   -3.0   |     |   78.0   |   74.4   |        -3.6         |   75.0   |   -3.0   |
> > | DeCLIP (ViT-B/32) | **78.7** |   70.8   |   -7.9   |   70.4   |   -8.3   |     |   81.9   |   79.2   |        -2.7         |   79.1   |   -2.8   |     | **82.1** | **80.2** |      **-1.9**       | **80.0** | **-2.1** |
> > | REPE (ViT-B/32)   |   78.5   | **70.9** | **-7.6** | **70.6** | **-7.9** |     | **82.3** | **79.8** |      **-2.5**       | **79.6** | **-2.7** |     |   79.0   |   76.4   |        -2.6         |   76.2   |   -2.8   |

---

> > > ### Author Response · Authors · 2022-11-18
> > > **Responses to Reviewer 8DmZ (Part 3)**
> > >
> > > **(A5 continued)** Besides, **our REPE is model-agnostic, making it orthogonal to the existing pre-training paradigms and parameter-tuning methods.**
> > >
> > > For example, we can combine it with DeCLIP to enhance its performance even more:
> > >
> > > | Model                    |          |          | CIFAR100 |          |          |
> > > |--------------------------|:--------:|:--------:|:--------:|:--------:|:--------:|
> > > |                          |  Acc-C   |  Acc-E   | $\Delta$ |  Acc-S   | $\Delta$ |
> > > | CLIP (ViT-B/32)          |   78.0   |   69.6   |   -8.4   |   68.9   |   -9.1   |
> > > | CLIP + REPE (ViT-B/32)   | **78.5** | **70.9** | **-7.6** | **70.6** | **-7.9** |
> > > | DeCLIP (ViT-B/32)        |   78.7   |   70.8   |   -7.9   |   70.4   |   -8.3   |
> > > | DeCLIP + REPE (ViT-B/32) | **79.8** | **72.9** | **-6.9** | **72.0** | **-7.8** |
> > >
> > > **To further improve performance, we can combine it with fine-tuning techniques like adapter-tuning** (also known as CLIP-adapter, please refer to Appendix I).
> > >
> > > | K-Shot |         Method        | CIFAR100 | ImageNet |
> > > |-------|---------------------|:--------:|:--------:|
> > > | 4     |     CLIP-Adatper    |   66.6   |    63.0      |
> > > | 4     | CLIP-Adapter + REPE | **67.5** |   **63.3**   |
> > > | 16    |     CLIP-Adapter   |   69.0   |    64.6     |
> > > | 16    | CLIP-Adapter + REPE | **69.8** |   **64.9**   |
> > >
> > > All these results demonstrate the superiority of our method.
> > >
> > > **We hope our response can address your concerns and we would appreciate it if you could reassess the paper.**

---

### Official Review · Reviewer_No3A · 2022-10-27

**Confidence:** 4
**Correctness:** 4
**Technical Novelty And Significance:** 2
**Empirical Novelty And Significance:** 2
**Recommendation:** 6

**Clarity, Quality, Novelty And Reproducibility:**

**Clarity**: The paper is easy to follow, with proper references to additional material in the appendices.

**Quality**: The work is strongly assertive of the issues with using CLIP-like models for open set recognition. Additionally, the experiments are extensively done to expose and formalize the openness-issue with extensibility and stability concepts. Additionally, the use of REPE kind of suggests the direction to improve for future models.

**Novelty**: It is an issue I am very confused with the given work. Novel in what sense? The paper does provide good insights by using their extensibility and stability analysis. This can be considered as novel direction to evaluate for making the CLIP-like models more ‘evaluable’.

**Strength And Weaknesses:**

**Strength**

1. The paper is one of the firsts to investigate the openness of CLIP-like models, with 2 evaluation measures: _extensibility_ and _stability_. And the paper proposes a method called REPE (Retrieval Enhanced Prompt Engineering) to tackle the two issues.

2. The experiments demonstrate that the CLIP-like models are indeed susceptible to fail in visual recognition, even with marginal difference in text features of different classes.

3. The experiments highlight that such models, which are decent zero-shot learners, do not scale well for larger vocabulary size.


**Weakness**

1. What is the significance of using REPE, when the incremental advantages do not seem visually (Fig.7) or numerically (Tab. 2) considerable?

2. In addition to retrieving the K nearest image captions and filtering out the class name to add visual semantics in REPE, is it possible to use the bottom-K captions and recompute the accuracies (C, E, S)?  The motivation of the paper is quite clear, however I wonder how much would the accuracies suffer if given the worst captions. The increase in accuracies being marginal, this thought occurred to me.

**Summary Of The Paper:**

The paper tackles three major issues with CLIP-like models: openness, extensibility and stability. The paper argues that CLIP-like models are difficult to evaluate since they are unconstrained in terms of the vocabulary use. Therefore the paper proposes to use an incremental evaluation perspective, called extensibility, to test the the models’ ability to learn new concepts.

The paper also suggests that the ambiguity in openness is not due to the failure to capture image-text similarity, but rather confusion among similar textual features. For this, the paper proposes to use a retrieval-augmented relevant texts to impose distinction between competing texts which the paper shows improves the extensibility and stability of CLIP without fine-tuning.

**Summary Of The Review:**

The paper does good job on highlighting and quantifying issue of open world recognition with CLIP-like models. However, there are other methods poised to be developed since it seems to be a known issue. Therefore, REPE giving marginal gains does not seem a strong base for future improvement. The experiments are sufficient and extensive for the given proposal.

Given many insights into the analysis with extensibility and stability, I believe that the paper has good potential and value to add to the community.

--------------------------------------------------------------------------------------
After reviewing the responses, I concur with other reviewers that the paper has low technical novelty. REPE is not very superior to other methods.

However, I still believe that the paper is good work and could serve as a nice direction for future work. I will keep my rating to above acceptance threshold.

---

> ### Author Response · Authors · 2022-11-19
> **Responses to Reviewer No3A**
>
> Thank you for the comments, and the recognition of our contribution. We provide the response to your questions as follows.
>
> > **Q1: What is the significance of using REPE, when the incremental advantages do not seem visually (Fig.7) or numerically (Tab. 2) considerable?**
>
> **A1:** We argue that improving the performance of zero-shot image classification is not trivial, and that **our performance improvement is substantial and comparable to powerful pre-training techniques like DeCLIP**:
>
> | Model | [ | | CIFAR100 | | | | ][ | | ImageNet (Entity13) | | | | ][ | | ImageNet (Living17) | | ] |
> |-------------------|:--------:|:--------:|:--------:|:--------:|:--------:|:---:|:--------:|:--------:|:-------------------:|:--------:|:--------:|:---:|:--------:|:--------:|:-------------------:|:--------:|:--------:|
> | | Acc-C | Acc-E | $\Delta$ | Acc-S | $\Delta$ | | Acc-C | Acc-E | $\Delta$ | Acc-S | $\Delta$ | | Acc-C | Acc-E | $\Delta$ | Acc-S | $\Delta$ |
> | CLIP (ViT-B/32) | 78.0 | 69.6 | -8.4 | 68.9 | -9.1 | | 80.8 | 78.0 | -2.8 | 77.8 | -3.0 | | 78.0 | 74.4 | -3.6 | 75.0 | -3.0 |
> | DeCLIP (ViT-B/32) | **78.7** | 70.8 | -7.9 | 70.4 | -8.3 | | 81.9 | 79.2 | -2.7 | 79.1 | -2.8 | | **82.1** | **80.2** | **-1.9** | **80.0** | **-2.1** |
> | REPE (ViT-B/32) | 78.5 | **70.9** | **-7.6** | **70.6** | **-7.9** | | **82.3** | **79.8** | **-2.5** | **79.6** | **-2.7** | | 79.0 | 76.4 | -2.6 | 76.2 | -2.8 |
>
> Besides, there are many advantages in our REPE:
>
> **First, REPE is model-agnostic, making it orthogonal to the existing pre-training paradigms and parameter-tuning methods.** For example, we can combine it with DeCLIP to enhance its performance even more:
>
> | Model | | | CIFAR100 | | |
> |--------------------------|:--------:|:--------:|:--------:|:--------:|:--------:|
> | | Acc-C | Acc-E | $\Delta$ | Acc-S | $\Delta$ |
> | CLIP (ViT-B/32) | 78.0 | 69.6 | -8.4 | 68.9 | -9.1 |
> | CLIP + REPE (ViT-B/32) | **78.5** | **70.9** | **-7.6** | **70.6** | **-7.9** |
> | DeCLIP (ViT-B/32) | 78.7 | 70.8 | -7.9 | 70.4 | -8.3 |
> | DeCLIP + REPE (ViT-B/32) | **79.8** | **72.9** | **-6.9** | **72.0** | **-7.8** |
>
> To further improve performance, we can combine it with fine-tuning techniques like adapter-tuning (also known as CLIP-adapter, please refer to Appendix I).
> | K-Shot | Method | CIFAR100 | ImageNet |
> |-------|---------------------|:--------:|:--------:|
> | 4 | CLIP-Adatper | 66.6 | 63.0 |
> | 4 | CLIP-Adapter + REPE | **67.5** | **63.3** |
> | 16 | CLIP-Adapter | 69.0 | 64.6 |
> | 16 | CLIP-Adapter + REPE | **69.8** | **64.9** |
>
> **Second, REPE is simple and non-parametric, which requires no fine-tuning. The cost of REPE is affordable.** Even though the feature pre-encoding and indexing process takes roughly two hours, it only needs to be done once. Once complete, we can quickly use FAISS to search the dataset in accordance with the image query in real time, for example, taking less than 5 ms to retrieve the top 100 pertinent picture-caption pairs.
>
> All these results demonstrate the superiority of our method.
>
> > **Q2: What’s the results of REPE if the most dissimilar captions are used?**
>
> **A2:** That’s an interesting point. We use the bottom-K captions (denotes as **REPE***) and recompute the accuracies as follows:
>
> | Model | [ | | CIFAR100 | | | | ][ | | ImageNet (Entity13) | | | | ][ | | ImageNet (Living17) | | ] |
> |------------------|:--------:|:--------:|:--------:|:--------:|:--------:|:---:|:--------:|:--------:|:-------------------:|:--------:|:--------:|:---:|:--------:|:--------:|:-------------------:|:--------:|:--------:|
> |                  |  Acc-C   |  Acc-E   | $\Delta$ |  Acc-S   | $\Delta$ |     |  Acc-C   |  Acc-E   |      $\Delta$       |  Acc-S   | $\Delta$ |     |  Acc-C   |  Acc-E   |      $\Delta$       |  Acc-S   | $\Delta$ |
> | CLIP (ViT-B/32)  |   78.0   |   69.6   |   -8.4   |   68.9   |   -9.1   |     |   80.8   |   78.0   |        -2.8         |   77.8   |   -3.0   |     |   78.0   |   74.4   |        -3.6         |   75.0   |   -3.0   |
> | REPE (ViT-B/32)  | **78.5** | **70.9** | **-7.6** | **70.6** | **-7.9** |     | **82.3** | **79.8** |      **-2.5**       | **79.6** | **-2.7** |     | **79.0** | **76.4** |      **-2.6**       | **76.2** | **-2.8** |
> | REPE* (ViT-B/32) |   71.2   |   63.5   |   -7.7   |   63.1   |   -8.1   |     |   77.3   |   74.6   |        -2.7         |   74.5   |   -2.8   |     |   73.9   |   71.1   |        -2.8         |   71.0   |   2.9    |
>
> The extensibility and stability of REPE* degrade by an average of 5.95% on all three datasets, which validates the motivation of our REPE.
>
> We hope we have addressed your concerns. Discussions are always open. Thank you!

---

### Author Response · Authors · 2022-11-19
**General Response**

We want to thank all the reviewers for their constructive comments. Specifically, we appreciate that:
1. All four reviewers concur that the openness issue we pointed out is a worthwhile and significant one. They recognize the value of our thorough and sufficient experiments as well as the depth and insight of our research.
2. Reviewers No3A appreciate that our REPE method suggests the direction to improve future models.
2. Reviewers No3A, 8DmZ, and c4Um agree that the paper is well-written and quite easy to follow

We have revised our paper accordingly and provided individual responses to each reviewer. The main changes to the paper are summarized as follows:
1. In Section 2.2, we emphasize the zero-shot setting in our evaluation, and clarify the advance of our evaluation protocol compared to the vanilla evaluation mechanism.
2. In Section 3.3, we describe the motivation of our REPE method in more detail and justify the effectiveness of REPE in addressing the openness issue.

---

### Decision · Program_Chairs · 2023-01-20

**Decision:**

Reject

**Justification For Why Not Higher Score:**

The reviewers agreed that the limitations of the paper do not warrant a higher score, and the one positive reviewer acknowledge the limitations as well. After the discussion, there seemed to be extensive agreement.

**Justification For Why Not Lower Score:**

N/A

**Metareview: Summary, Strengths And Weaknesses:**


 This paper explores the openness of CLIP, which has recently been used across a range of zero-shot and open-vocabulary settings, achieving seemingly impressive performance. Since understanding how truly "open" a model is can be difficult, the authors propose quantitative metrics characterizing the "extensibility" (i.e. effectiveness across the union of vocabularies) and "stability" (robustness to the introduction of non-target classes) of the model. Results across a number of datasets show that the vanilla CLIP method for zero-shot classification can suffer in these respects, and a prompt-based retrieval method is proposed to improve on these metrics. Results are demonstrated on CIFAR100 and ImageNet datasets.

  All of the authors appreciated the idea of this paper: The "openness" of CLIP is indeed something that is worthy of study and the authors' effort at quantifying this notion is commendable. However, beyond the results showing the limitations of CLIP, the authors had serious concerns including: 1) The incremental significance of the improvements over other state of art methods, 2) The novelty of the method and more importantly the tenuous link between the approach and the concepts described (extensibility/stability), and 3) Overall confusing flow for the paper including undefined terms. While the authors provided a rebuttal, including with some additional experiments, after extensive discussion by the reviewers there was a consensus that the current paper cannot be accepted as-is because the rebuttal does not address some of the fundamental issues in the paper.

  Indeed, the paper starts with an interesting proposition, attempt at quantifying openness, and demonstration of the limitations of the openness of CLIP. However, even besides the limited novelty and performance gain of the approach (which is not necessarily a reason by itself to reject a paper) the resulting approach is relatively detached from these finding. For example, the first part of the analysis is not even necessary for using the notion of intra-modal uniformity in the representations, which as cited has already been used to justify the improvement of models. As a result, the first and second parts of the paper are detached and lead to a poor organization for the paper. While the paper cannot be accepted in its current form, we recommend revising the paper to focus on either the first part (thorough analysis of CLIP's openness) or a method with more thorough and convincing improvements on CLIP's performance compared to the state of art.

**Summary Of Ac-Reviewer Meeting:**

N/A